# HOW STABLE IS THE NEXT TOKEN? A GEOMETRIC VIEW OF LLM PREDICTION STABILITY

**Deyuan Liu[1]**   **Zecheng Wang[1,2]**   **Zhanyue Qin[1]**   **Zhiying Tu[1]**   **Dianhui Chu[1]**   **Dianbo Sui[1*]**
[1]Harbin Institute of Technology   [2]Wechat AI

## ABSTRACT

Large Language Models (LLMs) exhibit impressive capabilities yet suffer from sensitivity to slight input context variations, hampering reliability. Conventional metrics like accuracy and perplexity fail to assess local prediction robustness, as normalized output probabilities can obscure the underlying resilience of an LLM's internal state to perturbations. We introduce the **Token Constraint Bound** ($\delta_{\text{TCB}}$), a novel metric that quantifies the maximum internal state perturbation an LLM can withstand before its dominant next-token prediction significantly changes. Intrinsically linked to output embedding space geometry, $\delta_{\text{TCB}}$ provides insights into the stability of the model's internal predictive commitment. Our experiments show $\delta_{\text{TCB}}$ correlates with effective prompt engineering and uncovers critical prediction instabilities missed by perplexity during in-context learning and text generation. $\delta_{\text{TCB}}$ offers a principled, complementary approach to analyze and potentially improve the contextual stability of LLM predictions.

## 1 INTRODUCTION

Large Language Models (LLMs), such as GPT-4 (OpenAI et al., 2023), LLaMA (Touvron et al., 2023; Dubey et al., 2024) and Gemini (Team et al., 2023), demonstrate remarkable capabilities, yet paradoxically exhibit striking sensitivity to contextual nuances. This brittleness manifests as substantial performance variations due to subtle modifications: accuracy can fluctuate by up to 76% from minor formatting changes (Sclar et al., 2023) or range from 54% to 93% based on example order (Zhao et al., 2021). Such variations stem from alterations in prompt phrasing (Razavi et al., 2025), example selection and ordering (Lu et al., 2021), or even basic formatting. Despite established scaling laws (Kaplan et al., 2020; Hoffmann et al., 2022) fueling impressive in-context learning (Brown et al., 2020; Dong et al., 2022; Wei et al., 2023), evidence indicates that increased model scale does not inherently confer enhanced robustness; larger models may even exhibit new sensitivities (Lu et al., 2021; Wei et al., 2023). This underscores the urgent need for robust stability metrics in modern AI evaluation, particularly for reliable deployment in mission-critical applications demanding consistent performance (Weidinger et al., 2021; Herrera-Poyatos et al., 2025).

Appraising contextual influence with precision is imperative, yet existing evaluation frameworks prove inadequate. Task accuracy yields only an aggregate performance view, overlooking the stability of individual predictions amid contextual shifts. Perplexity (Jelinek et al., 1977), though standard for sequence likelihood (Liang et al., 2022; Holtzman et al., 2021), conflates probabilities, thereby obscuring local dynamics essential for robustness. Moreover, it often neglects internal state geometry and fails to ensure internal stability even for high-probability tokens (Cohen-Inger et al., 2025). Crucially, the softmax normalization applied to derive output probabilities can mask a prediction's underlying stability; high probability can arise from relative normalization, not necessarily from a robust internal state. This implies that a high token probability offers no guarantee that the originating internal state $h$ is itself resilient to minor variations. Even as emerging metrics (Zhang et al., 2024; Tian et al., 2023; Geng et al., 2023) address confidence and calibrationchiefly by aligning probabilities with correctness likelihood (Tian et al., 2023)they do not directly gauge the robustness of a specific next-token prediction's dominant rank to perturbations in the internal representation $h$. A well-calibrated, high-confidence prediction may therefore belie an unstable equilibrium within the internal state (Liu et al., 2025). This gap in assessing the immediate predictive mechanism's stability against internal perturbations is the direct impetus for our central research question:

---

[*]Corresponding author.

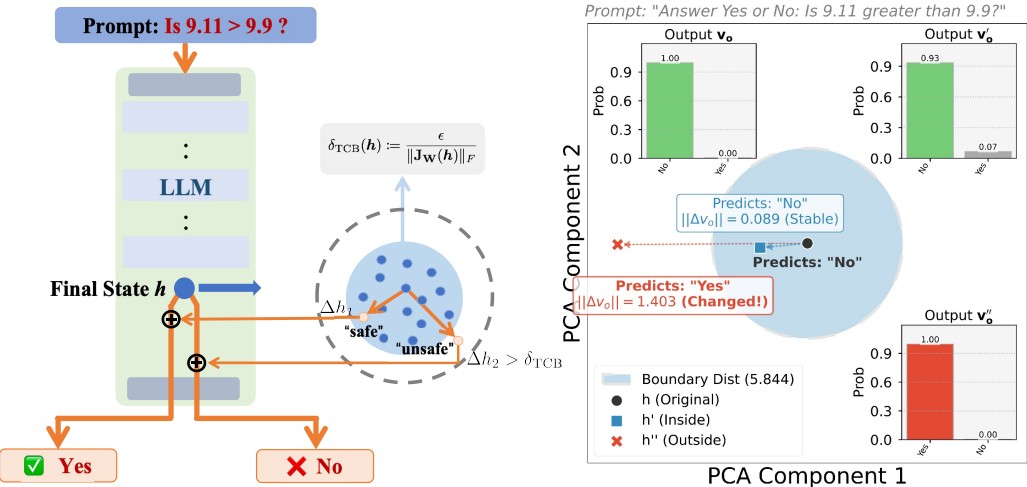

Figure 1: **The Token Constraint Bound ($\delta_{\text{TCB}}$) mechanism.** $\delta_{\text{TCB}}$ quantifies the maximum perturbation a model's internal state can withstand before the next-token prediction changes. (a) **Left panel** illustrates how a hidden state perturbation $\Delta h$ impacts the next token prediction. Small perturbations ($\Delta h_1$, implicitly within $\delta_{\text{TCB}}$ radius) may preserve the output, while larger ones ($\Delta h_2 > \delta_{\text{TCB}}$) can flip it (from "No" to "Yes"). $\delta_{\text{TCB}}$ bounds the perturbation size for stable output. (b) **Right panel** shows that the original hidden state $h$ and a perturbed state $h'$ inside a stability region predict "No". Another perturbation $h''$ outside the region flips the prediction to "Yes", demonstrating the practical consequence of exceeding the stability boundary.

> $\mathbb{Q}$**: How can we quantify the stability of an LLM's immediate prediction state, as induced by a specific prompt or context, against small internal variations?**

Addressing this question necessitates transcending aggregate performance metrics to develop measures specifically targeting the local robustness of prediction mechanisms. We must quantify how susceptible the next-token output distribution is to perturbations in the internal representation generated from the input contexta challenge at the intersection of representation stability and prediction reliability.

**Our approach.** We propose the **Token Constraint Bound** ($\delta_{\text{TCB}}$), a measure of this critical local stability. $\delta_{\text{TCB}}$ quantifies a "safety margin" around the internal state $h$ resulting from context processing: a larger $\delta_{\text{TCB}}$ means the models next-token prediction (particularly its top choice) withstands greater internal perturbations $\Delta h$ without significant change. It gauges the model's commitment to its current output ranking, given $h$. As explicated in Section 2 and depicted in Figure 1 , $\delta_{\text{TCB}}$ offers a direct measure of the output layer's robustness to hidden state variations. Therefore, a high $\delta_{\text{TCB}}$ signals a *stably* confident prediction state engendered by effective context.

We hypothesize that effective context, such as well-crafted prompts or informative ICL examples, not only guides models to correct answers but also induces a more *stable* internal state $h$, as reflected by higher $\delta_{\text{TCB}}$ values. This stability, signifying robust internal commitment to a predictive path, yields more reliable predictions. Consequently, $\delta_{\text{TCB}}$ offers a quantitative measure for context effectiveness beyond accuracy, serving as a proxy for the robustness of context-derived decision-making and complementing uncertainty metrics focused on "knowledge strength" (Ma et al., 2025).

Our experiments corroborate this. We show $\delta_{\text{TCB}}$ distinguishes prompt quality and exhibits distinct behaviors across confidence regimes, correlating with distributional flatness in uncertain cases and logit margins in high-confidence scenarios. Results confirm $\delta_{\text{TCB}}$'s sensitivity to output embedding geometry, its link to semantic content, and its ability to flag incipient instability during text generationdynamics perplexity overlooks.

**Our contributions are threefold:**

(a) We introduce and theoretically ground the Token Constraint Bound ($\delta_{\text{TCB}}$), a novel metric that measures the local robustness of an LLM's next-token prediction to internal state perturbations, and detail its practical computation ( Section 2 ).

(b) We derive an expression that intrinsically links $\delta_{\text{TCB}}$ to the geometric dispersion of output embeddings, thus identifying geometric underpinnings of prediction stability ( Section 3 ).

(c) Through empirical evaluation, we demonstrate $\delta_{\text{TCB}}$'s capacity to assess prompt effectiveness and showcase its application in refining both prompt engineering and ICL ( Section 4 ).

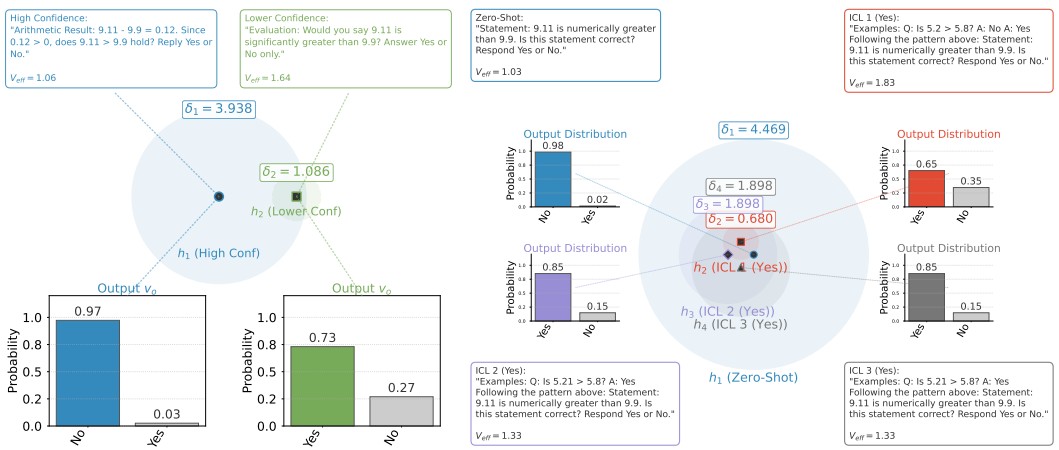

(a) Effect of prompt confidence on $\delta_{\text{TCB}}$.      (b) Effect of In-Context Learning on $\delta_{\text{TCB}}$.

Figure 2: $\delta_{\text{TCB}}$ **reflects context-induced prediction stability.** (a) Illustrates how prompts inducing higher prediction confidence (lower $\mathcal{V}_{\text{eff}}$, state $h_1$) lead to a significantly larger $\delta_{\text{TCB}}$ compared to prompts yielding lower confidence (higher $\mathcal{V}_{\text{eff}}$, state $h_2$). (b) Shows how In-Context Learning examples modify the hidden state and consequently the prediction and its stability. Adding examples can initially decrease stability while flipping the prediction, but consistent examples can increase stability for the target output.

## 2 PRELIMINARIES: UNDERSTANDING LLM PREDICTIONS AND STABILITY

This section establishes the foundational concepts for analyzing local output stability in LLMs. We begin by outlining the LLM output mechanism, then explore what it means for a prediction to be stable against internal variations. This leads to the introduction of the Jacobian matrix as a tool for quantifying sensitivity, and finally culminates in the formal definition of our $\delta_{\text{TCB}}$.

### 2.1 LANGUAGE MODEL OUTPUT AND DISTRIBUTION CONCENTRATION

Consider an LLM whose final layer computes a hidden state $h \in \mathbb{R}^d$. This state is linearly transformed by an output weight matrix $\mathbf{W} \in \mathbb{R}^{\mathcal{V} \times d}$ to produce logits $z = \mathbf{W}h$, where $\mathcal{V}$ is the vocabulary size. Each row $w_i^\top$ of $\mathbf{W}$ corresponds to the output embedding for token $i$. The probability distribution over the next token, $o \in \mathbb{R}^{\mathcal{V}}$, is obtained via the softmax function:

$$o = \text{softmax}(z), \quad \text{where } o_i = \frac{\exp(z_i)}{\sum_{j=1}^{\mathcal{V}} \exp(z_j)}. \tag{1}$$

This distribution satisfies $\sum_{i=1}^{\mathcal{V}} o_i = 1$ and $o_i \geq 0$. The model's prediction is typically the token $i^*$ maximizing $o_i$. A useful measure of the concentration of this distribution is the *effective vocabulary size* $\mathcal{V}_{\text{eff}}$:

$$\mathcal{V}_{\text{eff}}(o) := \frac{1}{\sum_{i=1}^{\mathcal{V}} o_i^2} = \frac{1}{\|o\|_2^2}. \tag{2}$$

$\mathcal{V}_{\text{eff}}$ ranges from 1 to $\mathcal{V}$, inversely relating to the $L_2$ norm squared of the probability vector. Our analysis hinges on understanding how the characteristics of this output vector $o$, including its concentration, relate to its *stability* when the context-derived hidden state $h$ undergoes small changes.

### 2.2 DEFINING STABILITY: WHAT DOES IT MEAN FOR A PREDICTION TO BE STABLE?

Our core objective is to understand the robustness of an LLM's next-token prediction. Specifically, we want to know: if the LLM's internal summary of the context (represented by the final hidden state $h$) changes slightly, how much does its next-token probability distribution $o$ change? Let $\Delta h \in \mathbb{R}^d$ represent a small internal "wobble" or perturbation to the hidden state $h$. Such a perturbation to the *model's internal representation of the context* could arise from minor input variations, noise in the computation, or other subtle disturbances. Let $o' = \text{softmax}(\mathbf{W}(h + \Delta h))$ be the perturbed output distribution. The resulting change in the prediction is $\Delta o = o' - o$.

The central question motivating our work, reiterated from the Introduction $\mathbb{Q}$, is how to quantify a "safety margin" for $h$: how large can the perturbation $\Delta h$ be before the change in the output $\Delta o$ becomes unacceptably large? Answering this requires a way to relate the magnitude of the internal

perturbation $\Delta\boldsymbol{h}$ to the magnitude of the resulting output change $\Delta\boldsymbol{o}$. This safety margin offers insights distinct from interpreting output probabilities $\boldsymbol{o}$ as direct measures of absolute confidence; instead, $\delta_{\mathrm{TCB}}$ focuses on the *local integrity and resilience of the current predictive mechanism* itself.

### 2.3 Quantifying the Impact of Perturbations: The Role of the Jacobian

To precisely relate changes in $\boldsymbol{h}$ to changes in $\boldsymbol{o}$, we utilize the concept of the Jacobian matrix. To first order, for small $\Delta\boldsymbol{h}$, the change in the output distribution $\Delta\boldsymbol{o}$ can be approximated linearly:

$$\Delta\boldsymbol{o} \approx \mathbf{J_W}(\boldsymbol{h})\Delta\boldsymbol{h}, \tag{3}$$

where $\mathbf{J_W}(\boldsymbol{h}) \in \mathbb{R}^{\mathcal{V} \times d}$ is the Jacobian matrix of the output probabilities $\boldsymbol{o}$ with respect to the hidden state $\boldsymbol{h}$. It is given by:

$$\mathbf{J_W}(\boldsymbol{h}) = \frac{\partial\boldsymbol{o}}{\partial\boldsymbol{h}} = \underbrace{\frac{\partial\boldsymbol{o}}{\partial\boldsymbol{z}}}_{\mathrm{diag}(\boldsymbol{o})-\boldsymbol{o}\boldsymbol{o}^\top} \underbrace{\frac{\partial\boldsymbol{z}}{\partial\boldsymbol{h}}}_{\mathbf{W}} = \left(\mathrm{diag}(\boldsymbol{o}) - \boldsymbol{o}\boldsymbol{o}^\top\right)\mathbf{W}. \tag{4}$$

The Jacobian $\mathbf{J_W}(\boldsymbol{h})$ essentially captures the sensitivity of each output probability $o_i$ to infinitesimal changes in each dimension of the hidden state $h_k$. Its entries are $\frac{\partial o_i}{\partial h_k}$. Note that the Jacobian depends on both the current output distribution $\boldsymbol{o}$ and the output weight matrix $\mathbf{W}$. To relate the overall magnitude of the state perturbation $\|\Delta\boldsymbol{h}\|_2$ to the overall magnitude of the output change $\|\Delta\boldsymbol{o}\|_2$, we use matrix norms. A standard inequality bounds the output change using the Jacobian's Frobenius norm:

$$\|\Delta\boldsymbol{o}\|_2 \leq \|\mathbf{J_W}(\boldsymbol{h})\|_F\|\Delta\boldsymbol{h}\|_2. \tag{5}$$

The Frobenius norm $\|\mathbf{J_W}(\boldsymbol{h})\|_F = \left(\sum_{i=1}^{\mathcal{V}}\sum_{k=1}^{d}\left(\frac{\partial o_i}{\partial h_k}\right)^2\right)^{1/2}$ provides a comprehensive, aggregate measure of the sensitivity of all output probabilities to all hidden state dimensions. A larger Frobenius norm indicates that, the output probabilities are more sensitive to changes in the hidden state.

### 2.4 The Token Constraint Bound ($\delta_{\mathrm{TCB}}$): Our Measure of Stability

We are interested in finding the maximum allowable perturbation radius $\|\Delta\boldsymbol{h}\|_2$ such that the resulting change in the output distribution, as measured by its $L_2$ norm $\|\Delta\boldsymbol{o}\|_2$, remains below a predefined small tolerance $\epsilon > 0$. That is, we impose the condition $\|\Delta\boldsymbol{o}\|_2 \leq \epsilon$. Using the bound from Eq. (5), we require $\|\mathbf{J_W}(\boldsymbol{h})\|_F\|\Delta\boldsymbol{h}\|_2 \leq \epsilon$. Rearranging for $\|\Delta\boldsymbol{h}\|_2$ gives us:

$$\|\Delta\boldsymbol{h}\|_2 \leq \frac{\epsilon}{\|\mathbf{J_W}(\boldsymbol{h})\|_F}. \tag{6}$$

This naturally motivates our core metric for local output stability, which we define as this upper bound on the perturbation norm:

---

**Definition 1 (Token Constraint Bound $\delta_{\mathrm{TCB}}$).** *Given the output weight matrix $\mathbf{W}$, hidden state $\boldsymbol{h}$, resulting output distribution $\boldsymbol{o} = softmax(\mathbf{W}\boldsymbol{h})$, and a tolerance $\epsilon > 0$ for the maximum $L_2$ change allowed in $\boldsymbol{o}$, the* Token Constraint Bound $\delta_{\mathrm{TCB}}$ *at state $\boldsymbol{h}$ is defined as:*

$$\delta_{\mathrm{TCB}}(\boldsymbol{h}) \coloneqq \frac{\epsilon}{\|\mathbf{J_W}(\boldsymbol{h})\|_F}. \tag{7}$$

*Here, $\mathbf{J_W}(\boldsymbol{h})$ is the Jacobian given by Eq. (4) and $\|\cdot\|_F$ denotes the Frobenius norm.*

---

The parameter $\epsilon$ is a dimensionless scale factor chosen by the user, representing the desired tolerance for output distribution change. Consequently, $\delta_{\mathrm{TCB}}(\boldsymbol{h})$ quantifies the $L_2$-norm radius of the largest hyper-sphere of perturbations $\Delta\boldsymbol{h}$ around the current hidden state $\boldsymbol{h}$ that, to a first-order approximation, guarantees the change in the output probability vector $\boldsymbol{o}$ remains within $\epsilon$. A larger $\delta_{\mathrm{TCB}}(\boldsymbol{h})$ signifies that the model's *current prediction state $\boldsymbol{o}$*, as induced by the context leading to $\boldsymbol{h}$, is intrinsically more robust to small internal variations ("wobbles") in this hidden state. Conversely, a smaller $\delta_{\mathrm{TCB}}(\boldsymbol{h})$ indicates that the prediction mechanism is more sensitive to such internal perturbations at this specific point. The crucial term $\|\mathbf{J_W}(\boldsymbol{h})\|_F^2$ in the denominator has a fundamental connection to the geometry of the output embeddings, which we explore in detail in Section 3 .

## 3 $\delta_{\text{TCB}}$ VIA OUTPUT EMBEDDING GEOMETRY

Def. 1 introduced the Token Constraint Bound ($\delta_{\text{TCB}}$) as a measure of local output stability. To unlock its full diagnostic power and understand its nuanced behavior, we now dissect its core component: the Frobenius norm of the output Jacobian, $\|\mathbf{J_W}(\boldsymbol{h})\|_F$. This section reveals that $\|\mathbf{J_W}(\boldsymbol{h})\|_F$, and consequently $\delta_{\text{TCB}}$, is deeply intertwined with the *geometric arrangement* of the model's output token embeddings $\boldsymbol{w}_i$ relative to the current prediction probabilities $\boldsymbol{o}$. Intuitively, a prediction is expected to be more stable if the leading token's embedding is well-isolated from competitors, or if the model is highly certain (peaked $\boldsymbol{o}$).

### 3.1 THE GEOMETRY OF OUTPUT: TOKEN EMBEDDINGS AND THEIR MEAN

Recall that the output weight matrix $\mathbf{W} \in \mathbb{R}^{\mathcal{V} \times d}$ contains the output embedding vector $\boldsymbol{w}_i^{\top}$ for each token $i$ as its rows. A key concept in understanding the geometric influence on stability is the probability-weighted mean embedding vector:

$$\boldsymbol{\mu_w}(\boldsymbol{h}) \coloneqq \sum_{j=1}^{\mathcal{V}} o_j \boldsymbol{w}_j = \mathbf{W}^{\top} \boldsymbol{o}. \tag{8}$$

Here, $\boldsymbol{\mu_w}(\boldsymbol{h}) \in \mathbb{R}^d$ represents the current probability-weighted locus within the embedding space. This mean vector, $\boldsymbol{\mu_w}(\boldsymbol{h})$, reflects the current probability-weighted locus within the embedding space, effectively representing the "center of mass" or the resultant directional influence of the entire output distribution on the embedding geometry.

### 3.2 DERIVING THE JACOBIAN NORM: CONNECTING SENSITIVITY TO EMBEDDING SPREAD

We now derive an exact analytical expression for the squared Frobenius norm of the output Jacobian, $\|\mathbf{J_W}(\boldsymbol{h})\|_F^2$, which is the crucial term determining $\delta_{\text{TCB}}$. Let $\{\boldsymbol{w}_i\}_{i=1}^{\mathcal{V}}$ be the output embedding vectors (rows of $\mathbf{W}$) and $\boldsymbol{\mu_w}(\boldsymbol{h})$ be the probability-weighted mean embedding as defined in Eq. (8).

> **Proposition 1 (Exact Squared Jacobian Norm Appendix I ).** *For a given output weight matrix* $\mathbf{W}$ *and hidden state* $\boldsymbol{h}$*, let* $\boldsymbol{o} = \text{softmax}(\mathbf{W}\boldsymbol{h})$ *be the output probability vector. The squared Frobenius norm of the output Jacobian* $\mathbf{J_W}(\boldsymbol{h}) = (\text{diag}(\boldsymbol{o}) - \boldsymbol{o}\boldsymbol{o}^{\top})\mathbf{W}$ *is exactly:*
>
> $$\|\mathbf{J_W}(\boldsymbol{h})\|_F^2 = \sum_{i=1}^{\mathcal{V}} o_i^2 \|\boldsymbol{w}_i - \boldsymbol{\mu_w}(\boldsymbol{h})\|_2^2. \tag{9}$$
>
> *This sum represents the squared Euclidean distances between each embedding* $\boldsymbol{w}_i$ *and the mean embedding* $\boldsymbol{\mu_w}(\boldsymbol{h})$*, weighted by the corresponding squared probability* $o_i^2$*.*

**Interpretation of the Formula.** Eq. (9) is pivotal. It states that the overall sensitivity of the output distribution to hidden state perturbations (as captured by $\|\mathbf{J_W}(\boldsymbol{h})\|_F^2$) is determined by how "spread out" the token embeddings $\boldsymbol{w}_i$ are from their probability-weighted mean $\boldsymbol{\mu_w}(\boldsymbol{h})$, with each squared distance $\|\boldsymbol{w}_i - \boldsymbol{\mu_w}(\boldsymbol{h})\|_2^2$ being amplified or diminished by the square of its token's probability $o_i^2$. The $o_i^2$ weighting is crucial:

- Embeddings for tokens with very low probability $o_i$ (and thus low current ""evidence" or ""belief" from the model) contribute minimally to the sum, even if geometrically distant from $\boldsymbol{\mu_w}$. The model effectively de-weights their geometric influence on stability at this state.

- Embeddings for high-probability tokens (carrying significant ""evidence") contribute substantially, particularly if they are far from $\boldsymbol{\mu_w}$. Their geometric influence on the Jacobian norm is quadratically emphasized by $o_i^2$.

This $o_i^2$ weighting distinguishes Eq. (9) from measures like the trace of the standard probability-weighted covariance matrix of embeddings. This distinction arises directly from the definition of the softmax Jacobian and is fundamental for correctly interpreting $\delta_{\text{TCB}}$ (see Appendix G ).

### 3.3 THE FULL FORM OF $\delta_{\text{TCB}}$ AND ITS GEOMETRIC MEANING

Substituting the exact squared Jacobian norm from Prop. 1 Eq. (9) into the definition of $\delta_{\text{TCB}}$ Eq. (7) yields its complete form:

$$\delta_{\text{TCB}}(\boldsymbol{h}) = \frac{\epsilon}{\sqrt{\sum_{i=1}^{\mathcal{V}} o_i^2 \|\boldsymbol{w}_i - \boldsymbol{\mu_w}(\boldsymbol{h})\|_2^2}}. \tag{10}$$

Table 1: **Pearson Correlations validating regime-dependent stability.** Shows strong positive Corr($\delta_{\text{TCB}}$, $\mathcal{V}_{\text{eff}}$) (**r=0.95**) in diverse prompts (DPD, $N = 309$, broad regime), indicating stability driven by flatness. In contrast, strong positive Corr($\delta_{\text{TCB}}$, $z_k - z_{j*}$) (**r=0.62**) emerges in high-confidence cases (Low-$\mathcal{V}_{\text{eff}}$ Targeted, LVD, $N = 360$), where Corr($\delta_{\text{TCB}}$, $\mathcal{V}_{\text{eff}}$) is negligible ($r = 0.08$), confirming stability relies on top-token separation when confidence is high. Metrics are the Token Constraint Bound ($\delta_{\text{TCB}}$), effective vocabulary size ($\mathcal{V}_{\text{eff}}$), and the logit margin between the top two tokens ($z_k - z_{j*}$).

| Dataset (N Samples) | Corr($\delta_{\text{TCB}}$, $\mathcal{V}_{\text{eff}}$) | Corr($\delta_{\text{TCB}}$, $z_k - z_{j*}$) | Corr($z_k - z_{j*}$, $\mathcal{V}_{\text{eff}}$) |
|---|---|---|---|
| Diverse Prompts (DPD, $N = 309$) | **0.95** (Strong +) | -0.40 (Moderate -) | -0.41 (Moderate -) |
| Low-$\mathcal{V}_{\text{eff}}$ Targeted (LVD, $N = 360$) | 0.08 (Near Zero) | **0.62** (Strong +) | -0.60 (Strong -) |

This equation provides a clear geometric interpretation: $\delta_{\text{TCB}}$ is inversely proportional to the square root of the $o_i^2$-weighted sum of squared Euclidean distances between each token embedding $\boldsymbol{w}_i$ and the probability-weighted mean embedding $\boldsymbol{\mu_w}(\boldsymbol{h})$. A larger $\delta_{\text{TCB}}$ indicates higher local robustness of the prediction generated from $\boldsymbol{h}$. This geometric dispersion, weighted by $o_i^2$, directly dictates the "safety radius" around $\boldsymbol{h}$, within which the output distribution $\boldsymbol{o}$ changes by at most $\epsilon$. Understanding this relationship is key to interpreting how context shapes $\delta_{\text{TCB}}$ and, by extension, prediction stability, as conceptually illustrated in Figure 1 and Figure 3 .

### 3.4 INTERPRETING STABILITY ACROSS PREDICTION REGIMES

The exact expression for $\|\mathbf{J_W}(h)\|_F^2$ in Eq. (9) and the visualization in Figure 3 elucidate how $\delta_{\text{TCB}}$ behaves under different prediction certainties.

**High-Confidence Regime (Low $\mathcal{V}_{\text{eff}}$, Peaked $o$).** When $\boldsymbol{o}$ is highly peaked on token $k$ ($o_k \to 1, \mathcal{V}_{\text{eff}} \to 1$), $\boldsymbol{\mu_w}(h) \to \boldsymbol{w}_k$, causing $\|\mathbf{J_W}(h)\|_F^2 \to 0$ and $\delta_{\text{TCB}} \to \infty$ ( Figure 3 a). Here, extreme certainty implies extreme stability. The sum approximates to $\sum_{j \neq k} o_j^2 \|\boldsymbol{w}_j - \boldsymbol{w}_k\|_2^2$ ( Appendix J ). Crucially, competitor probabilities $o_j$ (and thus $o_j^2$) are super-exponentially sensitive to the logit margin between the top-two candidates, $z_k - z_{j*} = z_{\text{top1}} - z_{\text{top2}}$. Larger $z_k - z_{j*}$ values drastically reduce $\|\mathbf{J_W}(h)\|_F^2$, boosting $\delta_{\text{TCB}}$. This underpins the empirical positive correlation between $\delta_{\text{TCB}}$ and $z_k - z_{j*}$ when $\mathcal{V}_{\text{eff}}$ is low ( Table 1 , Low-$\mathcal{V}_{\text{eff}}$). Distances $\|\boldsymbol{w}_j - \boldsymbol{w}_k\|_2^2$ further modulate this: more distant competitors require even smaller $o_j^2$ for the same stability.

**Uncertain Regime (Higher $\mathcal{V}_{\text{eff}}$, Flatter $o$).** When

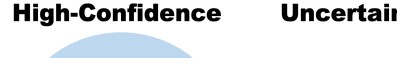

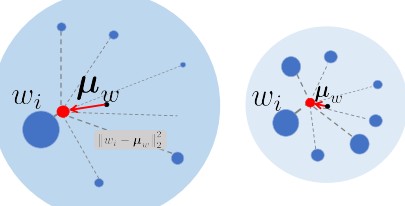

Figure 3: **Output Distribution Determines Geometric Stability.** The Token Constraint Bound ($\delta_{\text{TCB}}$) is a function of the geometric arrangement of embeddings. (a) **High Confidence:** Peaked distribution concentrates $\boldsymbol{\mu_w}(h)$ near the dominant embedding $\boldsymbol{w}_k$, minimizing the sum and maximizing $\delta_{\text{TCB}}$. (b) **Uncertainty:** Flatter distribution spreads $\boldsymbol{\mu_w}(h)$ among active embeddings, increasing the sum and reducing $\delta_{\text{TCB}}$. $\boldsymbol{\mu_w}(h)$, token embeddings $\boldsymbol{w}_i$.

probability is spread over multiple tokens (larger $\mathcal{V}_{\text{eff}}$, Figure 3 b), many $o_i$ are non-negligible. If these probable tokens' embeddings $\boldsymbol{w}_i$ are distant from $\boldsymbol{\mu_w}(h)$, their $\|\boldsymbol{w}_i - \boldsymbol{\mu_w}(h)\|_2^2$ terms increase $\|\mathbf{J_W}(h)\|_F^2$, reducing $\delta_{\text{TCB}}$. Crucially, however, a high $\mathcal{V}_{\text{eff}}$ does not *guarantee* low $\delta_{\text{TCB}}$: if high-probability embeddings $\{\boldsymbol{w}_i\}$ are geometrically *clustered* (and thus all near $\boldsymbol{\mu_w}(h)$), the $\|\boldsymbol{w}_i - \boldsymbol{\mu_w}(h)\|_2^2$ terms could be small despite significant $o_i^2$ values, potentially resulting in a larger $\delta_{\text{TCB}}$. This highlights geometry's primacy, validated by experiments where clustering embeddings (fixed $\boldsymbol{o}$) increases $\delta_{\text{TCB}}$. In this uncertain regime, under simplifying assumptions ( Appendix F , Appendix J ), approximations can suggest $\|\mathbf{J_W}(h)\|_F^2 \propto 1/\mathcal{V}_{\text{eff}}$. This implies $\delta_{\text{TCB}} \propto \sqrt{\mathcal{V}_{\text{eff}}}$, aligning with empirical correlations over diverse prompts ( Table 1 , Diverse Prompts), where overall distribution shape often dominates individual logit margins.

## 4 EXPERIMENTS

This section empirically substantiates the theoretical framework for $\delta_{\text{TCB}}$, focusing on its connection to output embedding geometry and its utility in LLM analysis. We address:
- $\delta_{\text{TCB}}$'s intrinsic properties, including sensitivity to output embedding geometry and correlations with standard metrics ($\mathcal{V}_{\text{eff}}$, $z_k - z_{j*}$) across confidence regimes ( Section 4.2 ).
- The role in diagnosing accuracy-stability conflicts and robust prompt engineering ( Section 4.3 ).
- How $\delta_{\text{TCB}}$ complements Perplexity (PPL) by assessing local prediction robustness ( Section D.3 ).

## 4.1 EXPERIMENTAL SETUP

• **Models.** Primary experiments utilize the LLAMA-3.1–8B model (Touvron et al., 2023). All computations were performed on NVIDIA RTX 4090 GPUs.

• **Datasets and Rationale.** **MMLU (Hendrycks et al., 2020):** Employed for broad validation and initial characterization due to its diverse subject matter. We typically sampled $N_{\text{init\_pool}} = 100$ questions from "test" splits of 3-5 reasoning-heavy subjects (e.g., `formal_logic`, `philosophy`) to assess general trends and robustness under varied content. **GSM8K (Cobbe et al., 2021):** Utilized for detailed intervention analysis and prompt optimization case studies. Its multi-step reasoning nature provides a fertile ground for examining how nuanced prompt changes affect both accuracy and internal stability. An initial pool of $N_{\text{init\_pool}} = 100$ questions from the "test" set was used for broader studies, with specific problems selected for deep dives. **DPD and LVD Datasets:** We synthesized two datasets for correlation analysis. The **Diverse Prompts Dataset (DPD)** contains prompts from a range of tasks, designed to elicit varied model confidence levels. The **Low-$\mathcal{V}_{\text{eff}}$ Targeted Dataset (LVD)** was created by modifying DPD prompts to generate high-confidence predictions.

• **Prompting Strategies.** *Zero-Shot:* Minimalist prompts for baseline correlation studies and initial conflict identification. *Few-Shot / Interventions:* A range of $k$-shot prompts ($k = 5$), ICL variations (e.g., algebraic vs. arithmetic focus, hyper-specific examples), and instructional prefixes were used in diagnostic analyses and optimization experiments.

• **Core Metrics.** At critical prediction points (e.g., just before generating the answer token for MMLU multiple-choice; before the first token of the final numerical answer in GSM8K): **Token Constraint Bound ($\delta_{\text{TCB}}(h)$):** Computed via Eq. (10). For all experiments, we set the tolerance parameter $\epsilon = 1.0$, which normalizes the metric. Since our analysis focuses on *relative changes* in stability, the specific value of $\epsilon$ is less critical than its consistency. Higher $\delta_{\text{TCB}}$ indicates greater internal state robustness. **Effective Vocabulary Size ($\mathcal{V}_{\text{eff}}(o)$):** From Eq. (2). Lower $\mathcal{V}_{\text{eff}}$ indicates a more peaked, confident distribution. **Logit Margin ($z_k - z_{j*}$):** Defined as $z_{\text{top1}} - z_{\text{top2}}$. Larger positive values suggest stronger discrimination for the top choice. **Task-Specific Accuracy (Acc):** Binary score based on ground truth. **Perplexity (PPL):** For local analysis in relation to $\delta_{\text{TCB}}$, we often refer to the negative log probability of the predicted token ($-\log o_{\text{predicted}}$). Further details on experimental configurations are in Appendix C .

## 4.2 INTRINSIC PROPERTIES AND VALIDATION OF $\delta_{\text{TCB}}$

**Validating Sensitivity to Output Embedding Geometry Objective:** To empirically confirm $\delta_{\text{TCB}}$'s direct dependence on output embedding geometry ($\mathbf{W}$), independent of the probability distribution ($o$). **Method Summary:** We synthetically manipulated $\mathbf{W}$ (clustering/dispersing competitor embeddings) while holding $h$ and $o$ (thus local PPL) constant for diverse MMLU prompts. $\delta_{\text{TCB}}$ was recalculated. **Results:** Table 2 shows the hypothesis $\delta_{\text{TCB}}(\mathbf{W}_{\text{cluster}}) > \delta_{\text{TCB}}(\mathbf{W}_{\text{orig}}) > \delta_{\text{TCB}}(\mathbf{W}_{\text{disperse}})$ held for **90% of prompts overall**, directly substantiating the geometric term in Eq. (10). This highlights that $\delta_{\text{TCB}}$ captures a dimension of stability tied to the embedding space that probability-only metrics would miss.

Table 2: **Simulation confirms $\delta_{\text{TCB}}$'s geometric sensitivity.** Percentage of prompts validating $\delta_{\text{cluster}} > \delta_{\text{orig}} > \delta_{\text{disperse}}$ when manipulating $\mathbf{W}$ ($K = 10$ competitors) while fixing $o$. The **effect robustly held (90% overall)**, confirming geometric influence distinct from probability shape.

| Prompt Category | Hypothesis Held |
|---|---|
| Low $\mathcal{V}_{\text{eff}}$ (< 20) | 95% |
| Medium $\mathcal{V}_{\text{eff}}$ (20-100) | 92% |
| High $\mathcal{V}_{\text{eff}}$ (> 100) | 80% |
| **Overall** | **90%** |

**Correlations Across Different Confidence Regimes  Objective:** To validate predicted shifts in $\delta_{\text{TCB}}$'s correlations with $\mathcal{V}_{\text{eff}}$ and $z_k - z_{j*}$ based on prediction confidence. **Method Summary:** Two MMLU zero-shot datasets: Diverse Prompts (DPD, $N = 309$) and Low-$\mathcal{V}_{\text{eff}}$ Targeted (LVD, $N = 360$). **Results:** Table 1 confirms the theorized regime dependence. In the DPD (broad regime), **Corr($\delta_{\text{TCB}}, \mathcal{V}_{\text{eff}}$) is** $0.95$, indicating stability is largely driven by overall distribution flatness. In stark contrast, for the LVD (high-confidence), Corr($\delta_{\text{TCB}}, \mathcal{V}_{\text{eff}}$) drops to a negligible $0.08$, while **Corr($\delta_{\text{TCB}}, z_k - z_{j*}$) becomes a strong** $0.62$. This shift empirically validates that when the model is confident, $\delta_{\text{TCB}}$ reflects the separation of the top token from its competitors rather than just the general peakedness of the distribution.

## 4.3 $\delta_{\text{TCB}}$ AS A DIAGNOSTIC TOOL FOR PROMPT ENGINEERING

$\delta_{\text{TCB}}$ uniquely identifies stability issues that accuracy or conventional confidence metrics ($\mathcal{V}_{\text{eff}}, z_k - z_{j*}$) may miss. Common *accuracy-stability conflict scenarios* include: (1) **Accurate but Unstable**:

Table 3: Combined Impact of $\delta_{\text{TCB}}$-Enhancement on Mean Metrics for Unperturbed and Perturbed Prompts (MMLU & GSM8K). Metrics are Accuracy (Acc), Token Constraint Bound ($\delta_{\text{TCB}}$), Effective Vocabulary Size ($\mathcal{V}_{\text{eff}}$), and Logit Margin ($z_k - z_{j*}$). Perturbed metrics include Accuracy Variance (AccVar$_{\text{pert}}$), Performance Drop Rate (PDR), and Worst-Case Accuracy (Acc$_{\text{worst}}$).

| | | Unperturbed Metrics | | | | Perturbed Metrics | | |
|---|---|---|---|---|---|---|---|---|
| Benchmark | Prompt Type | Acc | Avg. $\delta_{\text{TCB}}$ | Avg. $\mathcal{V}_{\text{eff}}$ | Avg. $z_k - z_{j*}$ | AccVar$_{\text{pert}}$ | PDR (%) | Acc$_{\text{worst}}$ |
| *Very Confident Questions (VCQ Set)* | | | | | | | | |
| MMLU | Baseline | 0.90 | 771.5 | 1.08 | 4.5 | 0.05 | 0.10 | 0.80 |
| | Enhanced | **0.95** | **1025.2** | **1.03** | **6.0** | **0.02** | **0.03** | **0.90** |
| GSM8K | Baseline | 0.85 | 2407.0 | 1.01 | 4.2 | 0.06 | 0.12 | 0.75 |
| | Enhanced | **0.92** | **4410.8** | **1.05** | **5.8** | **0.03** | **0.05** | **0.85** |
| *Ambiguous Questions (AQ Set)* | | | | | | | | |
| MMLU | Baseline | 0.40 | 1983.0 | 1.01 | 1.5 | 0.15 | 30 | 0.15 |
| | Enhanced | **0.70** | **2734.0** | **1.00** | **4.0** | **0.07** | **10** | **0.30** |
| GSM8K | Baseline | 0.35 | 3412.8 | 1.04 | 1.2 | 0.18 | 35 | 0.10 |
| | Enhanced | **0.65** | **6625.5** | **1.02** | **3.8** | **0.08** | **12** | **0.45** |

correct yet brittle predictions; (2) **Inaccurate but Stable**: robustly wrong predictions; (3) **Confident but Unstable**: high confidence indicators (e.g., $P(\text{top1})$ or $z_k - z_{j*}$) but low $\delta_{\text{TCB}}$; (4) **Uncertain but Stable**: flatter distribution but a resilient underlying state. These are elaborated with examples in Appendix D.1 .

**Case Study: Systematic Prompt Optimization Guided by $\delta_{\text{TCB}}$** **Objective:** To demonstrate using $\delta_{\text{TCB}}$ for guiding prompt engineering toward more robust solutions. **Method Summary:** We followed an iterative enhancement process on MMLU & GSM8K, detailed in Appendix C.4.2 . First, we ran baseline prompts over 3-5 random seeds to identify "Very Confident Questions" (VCQ; high and stable accuracy) and "Ambiguous Questions" (AQ; low/unstable accuracy or low $\delta_{\text{TCB}}$). Second, for AQ sets, we performed targeted prompt engineering, systematically refining components like ICL examples and instructional phrasing to co-optimize for both accuracy and $\delta_{\text{TCB}}$. Finally, we evaluated the enhanced prompts on unperturbed and perturbed data to measure gains in performance and robustness.

**Results:** Table 3 (illustrative of observed trends) shows that $\delta_{\text{TCB}}$-guided enhanced prompts achieve higher Acc and **significantly higher mean** $\delta_{\text{TCB}}$ (e.g., for the MMLU AQ set, from 1983.0 to **2734.0**). More critically, they exhibit **superior robustness to perturbations**, exemplified by lower Performance Drop Rate (PDR) (e.g., MMLU AQ set PDR: $30\% \rightarrow$ **10%**) and higher worst-case accuracy (Acc$_{\text{worst}}$) (e.g., MMLU AQ set Acc$_{\text{worst}}$: $15\% \rightarrow$ **30%**). This underscores that co-optimizing for $\delta_{\text{TCB}}$ yields more dependable LLM performance, particularly under minor contextual shifts. It is noteworthy that even for the Ambiguous Questions (AQ) set, the average $\mathcal{V}_{\text{eff}}$ remains low (cf. Table 3 ), suggesting that ambiguity in correctness does not necessarily correspond to low model confidence; the model can be confidently wrong.

To benchmark $\delta_{\text{TCB}}$-guided optimization, we compared it against a baseline strategy of perplexity-guided selection, where prompts are chosen to minimize the negative log-probability of the target answer. As shown in Table 4 , while perplexity-guidance improves accuracy, co-optimizing for $\delta_{\text{TCB}}$ yields more robust solutions with higher stability and better worst-case performance under perturbation.

Table 4: **Comparison of Prompt Optimization Strategies.** Co-optimizing for $\delta_{\text{TCB}}$ leads to more robust prompts compared to solely optimizing for perplexity (PPL), showing higher worst-case accuracy (Acc$_{\text{worst}}$) under perturbation.

| Optimization Strategy | Avg. Acc | Avg. $\delta_{\text{TCB}}$ ↑ | Avg. PPL ↓ | Acc$_{\text{worst}}$ |
|---|---|---|---|---|
| Baseline Prompt | 0.55 | 15.4 | 3.2 | 0.25 |
| PPL-Guided | 0.70 | 18.2 | **1.9** | 0.40 |
| $\delta_{\text{TCB}}$-Guided (Ours) | **0.72** | **35.8** | 2.4 | **0.65** |

### 4.3.1 DEEP DIVE: IMPACT OF PROMPT COMPONENT INTERACTIONS ON GSM8K

**Objective:** To showcase $\delta_{\text{TCB}}$'s fine-grained diagnostic capabilities in dissecting prompt component effects. **Method Summary:** On GSM8K problem gsm8k_811, we systematically varied ICLs, instructions, and question phrasing from an accurate baseline. **Results:** Table 5 highlights counter-intuitive accuracy-stability trade-offs. For instance, simply adding a clarifying phrase ("7 days",

Table 5: **Impact of Prompt Interventions on Accuracy and Stability Metrics for a GSM8K question.** Striking trade-offs between Acc and $\delta_{\text{TCB}}$ emerge. For instance, adding "(7 days)" (Row 2) tanks Acc to 0% but **boosts $\delta_{\text{TCB}}$ significantly (8.20 → 46.97)**, indicating a stable but incorrect state. Row 7 shows an extreme case: Zero-Shot with a strong instruction results in 0% Acc but **astronomical $\delta_{\text{TCB}} \approx 49$k**, epitomizing a "confidently and extremely stably wrong" prediction. Metrics shown are Accuracy (Acc), Token Constraint Bound ($\delta_{\text{TCB}}$), Effective Vocabulary Size ($\mathcal{V}_{\text{eff}}$), and top-2 logit margin ($z_k - z_{j*}$).

| Index | Intervention Description (gsm8k_811) | Acc (%) | $\delta_{\text{TCB}} \uparrow$ | $\mathcal{V}_{\text{eff}} \downarrow$ | $z_k - z_{j*} \uparrow$ |
|---|---|---|---|---|---|
| 1 | Baseline (New Algebraic ICLs, Original Question) | **100.0** | 8.20 | 1.54 | 3.25 |
| 2 | Clarified Q ("7 days") + New Alg. ICLs | 0.00 | **46.97** | 1.04 | 5.23 |
| 3 | Zero-shot CoT Instr. + Clarified Q + New Alg. ICLs | 0.00 | 10.95 | 1.44 | 2.09 |
| 4 | Role-Playing Instr. + Clarified Q + New Alg. ICLs | 0.00 | **62.14** | 1.03 | 5.98 |
| 5 | Algebraic Decomposition Instr. + Clarified Q + New Alg. ICLs | 0.00 | 10.38 | 1.33 | 3.62 |
| 6 | Hyper-Specific ICL + Alg. Decomp. Instr. + Clarified Q | 0.00 | **103.87** | 1.02 | 5.55 |
| 7 | Zero-Shot (No ICLs) + Alg. Decomp. Instr. + Clarified Q | 0.00 | **49450.23** | **1.00** | **11.29** |
| 8 | Formal Language Instr. + Clarified Q + New Alg. ICLs | 0.00 | **58.28** | 1.04 | 5.32 |

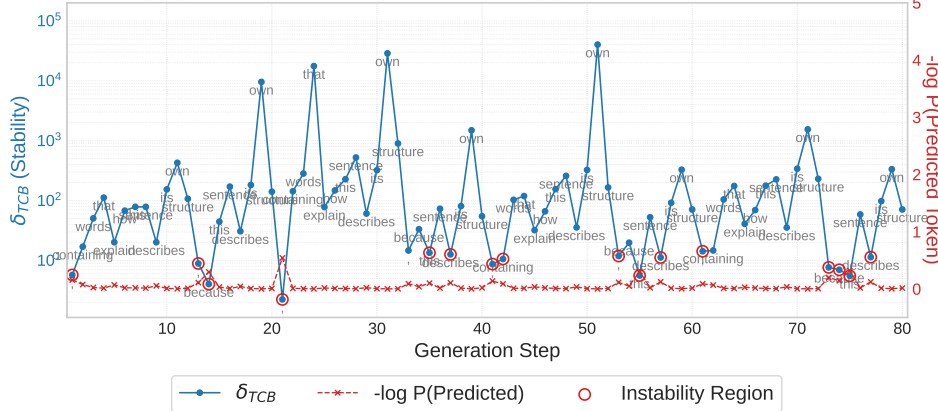

Figure 4: $\delta_{\text{TCB}}$ **dynamics vs. $P$(2nd best) during potentially repetitive generation.** Plot shows $\delta_{\text{TCB}}$ (blue, left y-axis) and $P(\text{2nd best})$ (green, right y-axis) versus generation step for LLAMA-3.1–8B. **Sharp dips in $\delta_{\text{TCB}}$** (e.g., around steps 5-10, 20-25) often correlate with **spikes in $P$(2nd best)**, indicating transient local instability not captured by average sequence PPL. Later, high, stable $\delta_{\text{TCB}}$ (e.g., steps 30+) can characterize a degenerate loop, showing robust commitment to the repetitive pattern.

Row 2) decimated accuracy (100% → 0%) but **boosted $\delta_{\text{TCB}}$ substantially (from 8.20 to 46.97)**, inducing a stable yet incorrect state. An even more extreme case is Row 7, where a zero-shot setup with a strong algebraic instruction yielded 0% accuracy but an **astronomical $\delta_{\text{TCB}} \approx 49$k** and perfect confidence ($\mathcal{V}_{\text{eff}} = 1.00$, $z_k - z_{j*} = 11.29$). This epitomizes an extremely "confidently and stably wrong" prediction. These findings underscore $\delta_{\text{TCB}}$'s capacity to uncover complex failure modes where models are robustly committed to erroneous reasoning pathsinsights that accuracy or standard confidence scores alone cannot provide.

## 5 CONCLUSION AND FUTURE WORK

To address Large Language Model sensitivity to input context variations, this paper introduces the Token Constraint Bound ($\delta_{\text{TCB}}$), a novel metric quantifying the local stability of next-token predictions against internal state perturbations. Intrinsically linked to the $o_i^2$-weighted geometric dispersion of output embeddings, $\delta_{\text{TCB}}$ offers a principled measure of an LLM's predictive commitment resilience. Our experiments demonstrate $\delta_{\text{TCB}}$'s utility in assessing prompt effectiveness and its ability to uncover critical prediction instabilities missed by perplexity, thus providing a valuable complementary tool for analyzing and potentially enhancing LLM contextual robustness. While these findings are promising, our current investigation primarily uses a specific LLM and a focused set of scenarios. Future work should expand this research across a broader spectrum of models, varying scales, and diverse application contexts to validate and generalize the utility of $\delta_{\text{TCB}}$. Exploring its application to understanding perturbations at intermediate layers could also yield deeper insights into representation robustness. Further investigation into $\delta_{\text{TCB}}$'s role in model editing, fine-tuning, and robustness against more structured attacks remains an important avenue.

ACKNOWLEDGEMENTS

This work is supported by the National Natural Science Foundation of China (Grant No. 62306087), the Key Research and Development Program of Shandong Provinc (Grant No. 2025CXPT077), the Natural Science Foundation of Shandong Province (Grant No. ZR2023QF154), the Research on Cognitive Processing Technologies for Multimodal Big Data in Policing Information Project (Grant No. 2024DXZD0004), and CCF-Tencent Rhino-Bird Open Research Fund (Grant No. CCF-Tencent RAGR20250105).

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

CONTENTS

## A    LLM USAGE STATEMENT

LLMs were used solely as auxiliary tools for paper polishing. They did not contribute to the generation of research ideas, the design of experiments, the development of methodologies, data analysis, or any substantive aspects of the research. All scientific content, conceptual contributions, and experimental results are entirely the work of the authors. The authors take full responsibility for the contents of this paper.

## B    RELATED WORK

Our work on the Token Constraint Bound ($\delta_{\text{TCB}}$) builds upon and differentiates itself from several lines of research in LLM evaluation, robustness, and interpretability.

**LLM Sensitivity and the Need for Robustness Metrics**    The pronounced sensitivity of LLMs to subtle input variations is well-documented. Studies have demonstrated substantial performance fluctuations arising from minor alterations in prompt phrasing (Razavi et al., 2025), example ordering and selection (Zhao et al., 2021; Lu et al., 2021), or even formatting details (Sclar et al., 2023). This brittleness (Marcus, 2020) highlights an urgent need for evaluation methods that go beyond aggregate task performance to assess the inherent stability of LLM predictions. Importantly, increased model scale, despite adherence to scaling laws (Kaplan et al., 2020; Hoffmann et al., 2022), does not invariably confer enhanced robustness to these nuanced changes (Lu et al., 2021; Wei et al., 2023), further motivating metrics like $\delta_{\text{TCB}}$ that focus on local stability. The imperative for reliable deployment in critical applications (Weidinger et al., 2021; Herrera-Poyatos et al., 2025; Harandizadeh et al., 2024) makes understanding and quantifying such instabilities paramount.

**Limitations of Conventional Evaluation Metrics**    Conventional metrics offer limited insight into local predictive robustness, a gap $\delta_{\text{TCB}}$ aims to address. Task accuracy, while a primary indicator, provides an aggregate view that can mask underlying prediction fragility due to contextual shifts (Zhao et al., 2021). Perplexity (Jelinek et al., 1977), though standard for assessing sequence likelihood (Liang et al., 2022; Holtzman et al., 2021), aggregates probabilities and can obscure local dynamics and the true stability of the internal state underpinning a prediction (Cohen-Inger et al., 2025). It may not reflect genuine model conviction, especially in cases of "surface form competition" (Holtzman et al., 2021) or when the softmax normalizes over poorly supported options. Emerging metrics targeting confidence and calibration (Zhang et al., 2024; Tian et al., 2023; Geng et al., 2023; Zhao et al., 2021) focus on aligning output probabilities with the likelihood of correctness (Tian et al., 2023). While valuable, these do not directly quantify the resilience of a specific prediction's dominant rank against perturbations in the model's internal representation $\boldsymbol{h}$ (Liu et al., 2025). A well-calibrated, high-confidence prediction might still stem from an internally fragile state. $\delta_{\text{TCB}}$ measures this internal representational stability directly.

**Internal State Dynamics and Representational Stability**    Our work intersects with research exploring LLM internal representations and their stability. Efforts to enhance robustness by, for example, aligning hidden states of perturbed instructions with original ones (Agrawal et al., 2025), implicitly underscore the importance of stable internal configurations. $\delta_{\text{TCB}}$ provides a direct, quantitative measure of this local stability specifically for the next-token prediction mechanism, assessing the "safety margin" or integrity of the current predictive commitment arising from $\boldsymbol{h}$. This connects to broader goals in Explainable AI (XAI) that seek to understand internal model workings (Mumuni & Mumuni, 2025), where $\delta_{\text{TCB}}$ can pinpoint internal decision points of high or low resilience.

**Prompt Engineering, In-Context Learning, and Stable State Induction**    $\delta_{\text{TCB}}$ is particularly relevant to analyzing the effectiveness of prompt engineering and In-Context Learning (Brown et al., 2020; Dong et al., 2022). Our central hypothesis posits that effective contextualization, such as through well-designed prompts or informative ICL examples, guides LLMs not only to correct answers but also to more *stable* internal states $\boldsymbol{h}$, reflected by higher $\delta_{\text{TCB}}$ values. This aligns with findings suggesting ICL's efficacy often stems from its ability to clarify task structure (Min et al., 2022). $\delta_{\text{TCB}}$ can thus serve as a quantitative tool to assess how effectively different contextual inputs induce robust internal commitments to a predictive path, complementing metrics focused on epistemic "knowledge strength" (Ma et al., 2025).

**Output Embedding Geometry and Predictive Stability**    A key theoretical underpinning of $\delta_{\text{TCB}}$, detailed in  Section 3 , is its intrinsic link to the geometric dispersion of output token embeddings. While the general importance of embedding space properties for model robustness is acknowledged

(Pang et al., 2025), $\delta_{\mathrm{TCB}}$ establishes a specific, analytical relationship between the geometry of the output embeddings (weighted by current prediction probabilities $o_i^2$) and the local stability of the next-token prediction against perturbations in $h$. This provides a mechanistic, geometric interpretation of local prediction stability, moving beyond probability-based analyses alone.

**Complementarity with Uncertainty Quantification (UQ)** $\delta_{\mathrm{TCB}}$ is positioned as complementary to, rather than a replacement for, existing Uncertainty Quantification (UQ) methods (Liu et al., 2025; Sychev et al., 2025). While UQ approaches typically aim to gauge a model's confidence or "knowledge strength" often based on output probabilities (e.g., entropy, prediction margins), $\delta_{\mathrm{TCB}}$ specifically assesses the robustness of the *predictive mechanism* itself to internal state fluctuations. A prediction can exhibit high output probability (high confidence by UQ standards) yet originate from an unstable internal state (low $\delta_{\mathrm{TCB}}$), indicating a "confidently unstable" scenario. Conversely, a moderately confident prediction might be highly stable. $\delta_{\mathrm{TCB}}$ thus offers a distinct perspective on reliability, focusing on the local integrity and resilience of the model's current predictive commitment rather than solely its expressed certainty.

**Perturbation-Based Robustness in Neural Networks and LLMs** Prior work on NN robustness has leveraged Jacobian norms and perturbations to quantify sensitivity and improve generalization. (Novak et al., 2018) empirically study sensitivity via the Frobenius norm of the input Jacobian, finding trained NNs more robust near training data and linking it to generalization gaps. Subsequent methods regularize this norm for adversarial robustness, e.g., (Hoffman et al., 2019) for classification margins. In LLMs, recent studies evaluate robustness to input perturbations like typos or rephrasing (Singh et al., 2024; Mumuni & Mumuni, 2025), with benchmarks like RUPBench (Wang & Zhao, 2024) showing larger models' resilience. $\delta_{\mathrm{TCB}}$ extends these by deriving an exact closed-form for the softmax-linear Jacobian norm, linking it to output embedding geometry, and repurposing it for contextual stability in ICL and prompt engineering—focusing on hidden state perturbations rather than inputs.

## C EXPERIMENTAL DETAILS

### C.1 DATASETS AND TASK-SPECIFIC SETUP

- **MMLU (Hendrycks et al., 2020):** For general validation (e.g., correlations in Table 1), questions were sampled from the 'test' splits of subjects like 'formal_logic', 'philosophy', 'abstract_algebra', 'moral_scenarios', and 'professional_law'. The standard multiple-choice format (A, B, C, D) was used. The critical prediction point for $\delta_{\mathrm{TCB}}$ calculation was immediately before the model generated the single token corresponding to its chosen letter (e.g., 'A', 'B', 'C', or 'D').
- **GSM8K (Cobbe et al., 2021):** For detailed intervention and prompt optimization studies, questions were sampled from the 'test' set. These are grade-school math word problems requiring multi-step reasoning. The critical prediction point for $\delta_{\mathrm{TCB}}$ was typically before the model generated the first token of the final numerical answer, after producing its chain-of-thought (CoT) reasoning. The final answer is usually identified by a pattern like "The final answer is \boxed{X}.".

### C.2 PROMPTING STRATEGIES AND EXAMPLES

#### C.2.1 MMLU PROMPTS

**Zero-Shot Multiple Choice:**

```
Question: {question_text}
Options:
A) {option_A_text}
B) {option_B_text}
C) {option_C_text}
D) {option_D_text}
Answer:
```

The model is expected to complete with 'A', 'B', 'C', or 'D'.

#### C.2.2 GSM8K PROMPTS (BASELINE AND INTERVENTIONS)

**Zero-Shot Chain-of-Thought (CoT):**

```
Question: {question_text}
Let's think step by step.
```

The model generates the reasoning steps and concludes with "The final answer is \boxed{X}."
**Few-Shot Chain-of-Thought (CoT) ($k = 5$ baseline for GSM8K):**

```
Question: {exemplar1_question_text}
Let's think step by step.
{exemplar1_CoT_solution}
The final answer is \boxed{{exemplar1_answer}}.
###
Question: {exemplar2_question_text}
Let's think step by step.
{exemplar2_CoT_solution}
The final answer is \boxed{{exemplar2_answer}}.
###
... (3 more exemplars) ...
###
Question: {current_question_text}
Let's think step by step.
```

Exemplars were typically drawn from the GSM8K 'train' set.

**Interventions for GSM8K problem `gsm8k_811` (as in  Table 5 ):** The base question `gsm8k_811` is: "Felix earns \$0.25 for each branch he trims from a tree. He trimmed branches from 12 trees. If he earned \$60 in total, what is the average number of branches he trimmed per tree?" (Correct Answer: 20)

- **Clarified Q ("7 days")**: The original question text was appended with: "(Felix works 7 days a week)." This clarification is irrelevant and misleading for this specific problem.
- **ICL Variations**:
    - *New Algebraic ICLs (Baseline ICLs for Table 12):* Exemplars were selected/written to emphasize setting up and solving algebraic equations (e.g., using variables like $x, y$).
    - *Hyper-Specific ICL:* An ICL example was crafted to be structurally almost identical to `gsm8k_811` (e.g., "John earns \$X per item. He processed Y items from Z batches. If he earned \$Total, what is the average items per batch?"), but with different numbers and context.
- **Instructional Prefixes**: These were typically inserted directly before "Let's think step by step." in a few-shot setup, or as the main instruction in a zero-shot setup.
    - *Zero-shot CoT Instr. (implied for Col 3 of Table 12 in context):* Simply "Let's think step by step." as the primary instruction for the new question.
    - *Role-Playing Instr.:* "You are a brilliant mathematician. Solve the following problem by showing your detailed work."
    - *Algebraic Decomposition Instr.:* "Decompose this problem algebraically. Define variables, set up equations, and solve them step-by-step to find the final answer."
    - *Formal Language Instr.:* "Use precise mathematical language and formal notation in your solution. Ensure each step is clearly justified."

For  Table 5 , "New Alg. ICLs" were used unless "Hyper-Specific ICL" or "No ICLs" (Zero-Shot) is specified.

### C.3 $\delta_{\mathrm{TCB}}$ CALCULATION AND PARAMETERS

The Token Constraint Bound $\delta_{\mathrm{TCB}}(\boldsymbol{h})$ was computed using Eq. (10) from the main text:

$$\delta_{\mathrm{TCB}}(\boldsymbol{h}) = \frac{\epsilon}{\sqrt{\sum_{i=1}^{\mathcal{V}} o_i^2 \left\| \boldsymbol{w}_i - \boldsymbol{\mu_w}(\boldsymbol{h}) \right\|_2^2}}$$

The tolerance parameter $\epsilon$ was set to 1.0 for all experiments. This choice normalizes $\delta_{\mathrm{TCB}}$ such that it represents the inverse of the Jacobian's Frobenius norm (scaled by $o_i^2$-weighted embedding variance). As stated in the main text, relative changes and comparative values of $\delta_{\mathrm{TCB}}$ are generally more informative than its absolute magnitude, which depends on $\epsilon$. The hidden state $\boldsymbol{h}$ was taken from the output of the final transformer layer, just before the unembedding layer.

## C.4 DETAILS OF SPECIFIC EXPERIMENTS

### C.4.1 GEOMETRY SENSITIVITY SIMULATION

For each input prompt: 1. The original hidden state $\boldsymbol{h}_{\text{orig}}$ and output probability distribution $\boldsymbol{o}_{\text{orig}}$ were obtained. $\delta_{\text{TCB}}(\boldsymbol{h}_{\text{orig}}, \mathbf{W}_{\text{orig}})$ was calculated. 2. The top $K = 10$ competitor tokens (tokens $j \neq \text{top1}$ with the highest $o_j$) were identified. 3. To create $\mathbf{W}_{\text{cluster}}$: For each competitor embedding $\boldsymbol{w}_j$ ($j \in K$ competitors), its new position was $\boldsymbol{w}_j^{\text{cluster}} = \boldsymbol{w}_j^{\text{orig}} + \alpha(\boldsymbol{w}_{\text{top1}}^{\text{orig}} - \boldsymbol{w}_j^{\text{orig}})$, where $\alpha = 0.5$ (moving it halfway towards the top-1 token's embedding). Embeddings of other tokens remained unchanged. $\delta_{\text{TCB}}(\boldsymbol{h}_{\text{orig}}, \mathbf{W}_{\text{cluster}})$ was calculated using the original $\boldsymbol{h}_{\text{orig}}$ and $\boldsymbol{o}_{\text{orig}}$. 4. To create $\mathbf{W}_{\text{disperse}}$: For each competitor embedding $\boldsymbol{w}_j$, its new position was $\boldsymbol{w}_j^{\text{disperse}} = \boldsymbol{w}_j^{\text{orig}} - \beta(\boldsymbol{w}_{\text{top1}}^{\text{orig}} - \boldsymbol{w}_j^{\text{orig}})$, where $\beta = 0.5$ (moving it away from the top-1 token's embedding along the same line, by half the original distance). $\delta_{\text{TCB}}(\boldsymbol{h}_{\text{orig}}, \mathbf{W}_{\text{disperse}})$ was calculated. The hypothesis $\delta_{\text{TCB}}(\mathbf{W}_{\text{cluster}}) > \delta_{\text{TCB}}(\mathbf{W}_{\text{orig}}) > \delta_{\text{TCB}}(\mathbf{W}_{\text{disperse}})$ was then checked.

### C.4.2 SYSTEMATIC PROMPT OPTIMIZATION PROTOCOL

**1. Baseline Characterization:** A set of questions (e.g., 50-100 from MMLU/GSM8K) was run with a baseline prompt (e.g., zero-shot for MMLU, 5-shot CoT for GSM8K) across multiple random seeds (e.g., $S = 3$ or $S = 5$) for ICL selection or minor phrasing variants to get Acc statistics. Metrics collected per question: Mean Accuracy, Accuracy Variance (across seeds), Mean $\delta_{\text{TCB}}$ (at critical token, averaged over seeds if multiple correct paths), $\delta_{\text{TCB}}$ Variance. **VCQ (Very Confident Questions)** were defined as those with, e.g., Mean Acc $\geq 0.9$, Acc Var $\leq 0.05$, Mean $\delta_{\text{TCB}} > T_H$, $\delta_{\text{TCB}}$ Var $< V_L$. **AQ (Ambiguous Questions)** were defined as those with, e.g., Mean Acc $< 0.6$, or Acc Var $> 0.15$, or Mean $\delta_{\text{TCB}} < T_L$, or $\delta_{\text{TCB}}$ Var $> V_H$. Thresholds $(T_H, T_L, V_L, V_H)$ were set empirically based on observed distributions.

**2.Targeted Prompt Enhancement:** For AQ questions, prompt engineering efforts focused on increasing both Acc and $\delta_{\text{TCB}}$. Techniques included: Refining ICL examples (e.g., ensuring CoT steps are clearer, more analogous to the target problem structure, varying reasoning styles). Modifying instructional phrases (e.g., adding "Be very careful with calculations," "Explain your reasoning clearly"). Switching prompting strategy (e.g., from 5-shot to 2-shot with very high-quality examples, or adding a self-reflection step). For MMLU, this could involve adding a directive like "Choose the best option and explain why." For VCQ questions, efforts might focus on further increasing $\delta_{\text{TCB}}$ if it wasn't already maximal, or ensuring robustness (see below).

**2.Evaluation (Unperturbed and Perturbed):** Enhanced prompts were evaluated on the same metrics. Robustness was tested using perturbed inputs (see Appendix C.4.3). Metrics like Accuracy Variance on perturbed inputs (AccVar$_{\text{pert}}$), Performance Drop Rate (PDR), and Worst-Case Accuracy (Acc$_{\text{worst}}$) under perturbation were key.

### C.4.3 PERTURBATION STRATEGIES FOR ROBUSTNESS EVALUATION

Perturbations were designed to be plausible minor variations:
- **Syntactic Perturbations (applied to the question text):**
    - Paraphrasing: Using a pre-trained paraphrasing model (e.g., T5-based) to generate 2-3 variants of the question.
    - Reordering: For questions with multiple clauses/conditions, reordering them if semantically permissible.
    - Synonym Replacement: Replacing 1-2 keywords with close synonyms.
- **Semantic Perturbations (primarily for ICL / Few-Shot setups):**
    - ICL Example Reordering: Changing the order of the few-shot examples in the prompt.
    - ICL Example Replacement: Replacing one of the $k$ examples with another valid but perhaps slightly less similar or slightly lower-quality example from the training set.
    - Adding a minor distractor sentence to the prompt context.

The goal was not to make the task unsolvable but to test sensitivity to typical input variations.

### C.4.4 TEXT GENERATION DYNAMICS SETUP

The prompt used to induce potentially repetitive behavior was:

```
System: Repeat the following word exactly five times: 'banana'.
After repeating it five times, say 'Task finished.'.
```

```
User: Okay, I understand. I will now repeat the word 'banana' five times
and then say 'Task finished.'
Assistant:
```

The model (LLAMA-3.1–8B) was allowed to generate tokens greedily. $\delta_{\text{TCB}}$, $P(\text{top1})$, $P(\text{2nd best})$, and $\mathcal{V}_{\text{eff}}$ were calculated at each token generation step. The figure typically shows dynamics if the model continues beyond the explicit instruction, e.g., by getting stuck in a loop of "banana" or "Task finished." or exhibiting other non-ideal behaviors. The "dips" in $\delta_{\text{TCB}}$ often occur at points where the model is less certain about continuing a pattern versus breaking out or switching to another token.

## D    FURTHER DISCUSSION AND EXAMPLES

### D.1    ELABORATION ON ACCURACY-STABILITY CONFLICT SCENARIOS

The main text ( Section 4.3 ) mentioned four key conflict scenarios. Here are more detailed conceptual examples:

**1.Accurate but Unstable (High Acc, Low $\delta_{\text{TCB}}$):**    *Scenario:* The model correctly answers a complex reasoning question, but its internal state $\boldsymbol{h}$ is near a decision boundary where a slight internal "wobble" could have led to a different (incorrect) reasoning path and answer. *Example (MMLU-like):* Q: "Which of these legal principles is most directly violated by ex post facto laws?" Correct Answer: "Nulla poena sine lege". Model predicts "Nulla poena sine lege" (Acc=1). However, its $\delta_{\text{TCB}}$ is low (e.g., 2.1). This might be because the embedding for another plausible but incorrect principle like "Stare decisis" is geometrically positioned such that a small perturbation to $\boldsymbol{h}$ could shift the logits sufficiently to make "Stare decisis" the top choice. The correct prediction is thus brittle.

**2.Inaccurate but Stable (Low Acc, High $\delta_{\text{TCB}}$):**    *Scenario:* The model makes a clear error, often due to a misinterpretation or flawed reasoning pattern, but it is very robustly committed to this error. *Example (GSM8K-like, from Table 5 , Row 2):* Felix problem with the misleading "7 days" clarification. The model incorrectly calculates the answer (Acc=0) but does so with high $\delta_{\text{TCB}}$ (46.97). It has latched onto a stable, but flawed, interpretation/procedure.

**3.Confident but Unstable (High $P(\text{top1})/z_k - z_{j^*}$, but Low $\delta_{\text{TCB}}$):**    *Scenario:* The output probability for the top token is high, and/or the logit margin to the next token is large, suggesting strong confidence. However, the internal state supporting this is not robust. *Example:* Prompt: "What is the primary ingredient in concrete?" Model predicts "Cement" with $P(\text{Cement}) = 0.95$ and a large logit margin $z_k - z_{j^*} = 5.0$. Superficially, this looks very confident. However, $\delta_{\text{TCB}}$ is low (e.g., 1.8). This could occur if: The embedding $\boldsymbol{w}_{\text{Cement}}$ is relatively far from the mean embedding $\boldsymbol{\mu_w}(\boldsymbol{h})$ (making the $o_{\text{Cement}}^2 \|\boldsymbol{w}_{\text{Cement}} - \boldsymbol{\mu_w}(\boldsymbol{h})\|_2^2$ term large despite $o_{\text{Cement}}^2$ being large too, if the distance is very large). Or, many other low-probability competitor embeddings $\{\boldsymbol{w}_j\}$ are clustered very tightly around $\boldsymbol{\mu_w}(\boldsymbol{h})$, leading to many small $o_j^2 \|\boldsymbol{w}_j - \boldsymbol{\mu_w}(\boldsymbol{h})\|_2^2$ terms whose sum is significant, contributing to a large Jacobian norm. The high probability $P(\text{Cement})$ might hide an underlying geometric configuration that is sensitive to perturbation.

**4.Uncertain but Stable (Low $P(\text{top1})/z_k - z_{j^*}$, High $\mathcal{V}_{\text{eff}}$, but High $\delta_{\text{TCB}}$):**    *Scenario:* The model's output distribution is relatively flat, indicating uncertainty among several top choices. Yet, the internal state representing this uncertainty is stable. *Example:* Prompt: "Which of these is a common pet: A) Dog, B) Tiger, C) Whale, D) Ant". Model output: $P(\text{Dog}) = 0.5$, $P(\text{Ant}) = 0.3$ (perhaps due to "common"), $P(\text{Tiger}) = 0.1$, $P(\text{Whale}) = 0.1$. $\mathcal{V}_{\text{eff}}$ is relatively high. However, $\delta_{\text{TCB}}$ could be high if the embeddings $\boldsymbol{w}_{\text{Dog}}$ and $\boldsymbol{w}_{\text{Ant}}$ are very close to each other (and thus to $\boldsymbol{\mu_w}(\boldsymbol{h})$ if they dominate), while $\boldsymbol{w}_{\text{Tiger}}$ and $\boldsymbol{w}_{\text{Whale}}$ are far away. The model is stably "stuck" deciding between "Dog" and "Ant", but it's not about to suddenly jump to "Tiger". The uncertainty itself has a stable geometric basis.

### D.2    DETAILED INTERPRETATION OF GSM8K INTERVENTION ANALYSIS

This provides a row-by-row interpretation of the results for GSM8K question `gsm8k_811` shown in Table 5 of the main paper.

- **Row 1: Baseline (New Algebraic ICLs, Original Question)** *Metrics:* Acc=100%, $\delta_{\text{TCB}} = 8.20$, $\mathcal{V}_{\text{eff}} = 1.54$, $z_k - z_{j^*} = 3.25$. *Interpretation:* The baseline prompt with algebraic ICLs successfully solves the problem. The stability ($\delta_{\text{TCB}} = 8.20$) is moderate, indicating a reasonably

robust correct prediction but with potential for improvement. $\mathcal{V}_{\text{eff}}$ and $z_k - z_{j^*}$ reflect good confidence.

- **Row 2: Clarified Q ("7 days") + New Alg. ICLs** *Metrics:* Acc=0%, $\delta_{\text{TCB}} = 46.97$, $\mathcal{V}_{\text{eff}} = 1.04$, $z_k - z_{j^*} = 5.23$. *Interpretation:* Adding the misleading "7 days" clarification breaks accuracy completely. Crucially, $\delta_{\text{TCB}}$ *increases dramatically*. This indicates the model has latched onto an incorrect interpretation or reasoning path due to the "7 days" phrase, and this incorrect path is highly stable. The high confidence metrics ($\mathcal{V}_{\text{eff}} \approx 1$, $z_k - z_{j^*}$ high) support this: it's "confidently and stably wrong."

- **Row 3: Zero-shot CoT Instr. + Clarified Q + New Alg. ICLs** *Metrics:* Acc=0%, $\delta_{\text{TCB}} = 10.95$, $\mathcal{V}_{\text{eff}} = 1.44$, $z_k - z_{j^*} = 2.09$. *Interpretation:* The standard "Let's think step by step" instruction does not fix the error caused by "7 days". The stability ($\delta_{\text{TCB}} = 10.95$) is higher than the original baseline (Row 1) but much lower than the "stably wrong" state in Row 2. This suggests the CoT instruction interacts with the misleading clarification and ICLs to produce a state that is still incorrect but less internally committed/stable than Row 2.

- **Row 4: Role-Playing Instr. + Clarified Q + New Alg. ICLs** *Metrics:* Acc=0%, $\delta_{\text{TCB}} = 62.14$, $\mathcal{V}_{\text{eff}} = 1.03$, $z_k - z_{j^*} = 5.98$. *Interpretation:* Accuracy remains 0. The Role-Playing instruction ("You are a brilliant mathematician...") leads to an even higher $\delta_{\text{TCB}}$ than Row 2, suggesting this type of instruction might encourage the model to commit more strongly to a particular reasoning path, even if that path is flawed due to other elements like the "7 days" clarification. Again, very high confidence in the wrong answer.

- **Row 5: Algebraic Decomposition Instr. + Clarified Q + New Alg. ICLs** *Metrics:* Acc=0%, $\delta_{\text{TCB}} = 10.38$, $\mathcal{V}_{\text{eff}} = 1.33$, $z_k - z_{j^*} = 3.62$. *Interpretation:* Similar to Row 3, this more specific algebraic instruction fails to correct the error. The stability is low, suggesting a conflict: the instruction pushes for algebraic rigor, but the "7 days" phrase may prevent a coherent algebraic formulation of the (misinterpreted) problem, leading to an unstable incorrect state.

- **Row 6: Hyper-Specific ICL + Alg. Decomp. Instr. + Clarified Q** *Metrics:* Acc=0%, $\delta_{\text{TCB}} = 103.87$, $\mathcal{V}_{\text{eff}} = 1.02$, $z_k - z_{j^*} = 5.55$. *Interpretation:* Even with a perfectly analogous ICL example and an algebraic instruction, the "7 days" clarification persists in causing an error. The $\delta_{\text{TCB}}$ is extremely high. This implies the model rigidly follows the structure of the hyper-specific ICL. If the "7 days" clarification leads to a consistent misapplication of that structure (e.g., always multiplying by 7 at a certain step because it's done in the (misinterpreted) exemplar logic), the result is a very stable, confident, but incorrect prediction.

- **Row 7: Zero-Shot (No ICLs) + Alg. Decomp. Instr. + Clarified Q** *Metrics:* Acc=0%, $\delta_{\text{TCB}} \approx 49450$, $\mathcal{V}_{\text{eff}} = 1.00$, $z_k - z_{j^*} = 11.29$. *Interpretation:* This is the most striking result. Removing ICLs (which might have conflicting signals) and relying solely on the strong "Algebraic Decomposition" instruction in the presence of the "7 days" clarification leads to an astronomically high $\delta_{\text{TCB}}$ and perfect confidence metrics, yet 0% accuracy. The model is utterly convinced by its (flawed) algebraic decomposition of the misinterpreted problem. This is the epitome of a "confidently and extremely stably wrong" state.

- **Row 8: Formal Language Instr. + Clarified Q + New Alg. ICLs** *Metrics:* Acc=0%, $\delta_{\text{TCB}} = 58.28$, $\mathcal{V}_{\text{eff}} = 1.04$, $z_k - z_{j^*} = 5.32$. *Interpretation:* Similar to the Role-Playing instruction (Row 4), asking for formal language seems to stabilize the incorrect reasoning path derived from the "7 days" clarification and algebraic ICLs, leading to high $\delta_{\text{TCB}}$ and confidence in the error.

This detailed analysis demonstrates $\delta_{\text{TCB}}$'s power in revealing how different prompt components interact to affect not just accuracy, but the internal stability and commitment of the model to its predictions, whether correct or incorrect.

### D.3    DIFFERENTIATING $\delta_{\text{TCB}}$ FROM PERPLEXITY (PPL)

$\delta_{\text{TCB}}$ assesses internal state robustness, a quality distinct from PPL's measure of sequence likelihood or token-level prediction confidence. Key differentiators include: **Robustness of (Potentially Erroneous) Confident Predictions:** PPL (or low token probability) flags incorrect predictions (if ground truth is known) but doesn't indicate if the model is *stably committed* to an error. $\delta_{\text{TCB}}$ quantifies this "stable incorrectness" (e.g., Table 5, Row 7, where $\delta_{\text{TCB}}$ is exceptionally high for an incorrect answer). **Sensitivity to Output Embedding Geometry:** PPL is solely a function of probabilities and thus blind to the geometric arrangement of output embeddings, which critically impacts $\delta_{\text{TCB}}$ and the true local stability of the prediction (cf. Section 4.2 and Eq. (10)). **Detection of Local Instabilities During Text Generation:** Sequence-level PPL averages token likelihoods,

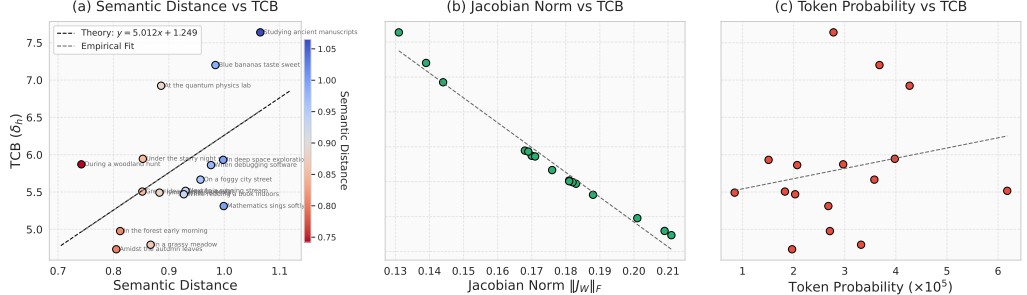

Figure 5: **Relationship between Prefix Semantics, TCB, and Model Internals.** Visual analysis based on 16 semantically varied prefixes targeting the same continuation. (a) $\delta_{\mathrm{TCB}}$ versus semantic distance shows a strong positive linear correlation ($R^2 = 0.91$). Point colors map to semantic distance (see colorbar), and annotations identify prefixes. The empirical fit (gray dashed line) closely matches the data, aligning well with the theoretical prediction (black dashed line, Eq. (11)). (b) $\delta_{\mathrm{TCB}}$ versus the Frobenius norm of the final layer Jacobian ($\|J_{\mathbf{W}}\|_F$) exhibits a moderate negative correlation. (c) $\delta_{\mathrm{TCB}}$ versus the probability of the first token in the continuation shows a less distinct relationship for this set of prefixes.

potentially masking transient points of high internal instability that can precede errors or degenerate loops. Figure 4 illustrates $\delta_{\mathrm{TCB}}$ capturing **sharp dips in** $\delta_{\mathrm{TCB}}$ (often correlating with **spikes in** $P(\textbf{2nd best token})$). These crucial local dynamics, indicative of the model teetering between alternatives, are often smoothed over by aggregate PPL but are vital for understanding generation failures.

## E QUANTIFYING SEMANTIC CONTEXT INFLUENCE.

Effective prompts often rely on precise semantic alignment. We investigated if $\delta_{\mathrm{TCB}}$ quantitatively reflects this. Using 16 prefixes with varying semantic distances (cosine distance of embeddings) to a target prefix, we measured the $\delta_{\mathrm{TCB}}$ of the first generated token. Figure 5 reveals a strong positive linear correlation ($R^2 = 0.91$) between semantic distance and $\delta_{\mathrm{TCB}}$. The empirical fit:

$$\delta_{\mathrm{TCB}} = 5.012 \cdot \mathrm{dist} + 1.249 \tag{11}$$

closely matches the data and theoretical predictions derived from sensitivity analysis. This shows $\delta_{\mathrm{TCB}}$ provides a quantitative measure of how semantic (mis)alignment in the prompt impacts the stability of the resulting predictive state. Figure 5 also shows the expected negative correlation between $\delta_{\mathrm{TCB}}$ and the Jacobian norm ($\|\mathbf{J}_{\mathbf{W}}\|_F$). This ability to quantify semantic influence further highlights $\delta_{\mathrm{TCB}}$'s utility for fine-grained prompt analysis and optimization.

## F STATISTICAL DERIVATION OF $\delta_{\mathrm{TCB}}$ APPROXIMATION

In this section, we provide a detailed derivation of a *statistical approximation* for the Token Constraint Bound ($\delta_{\mathrm{TCB}}$). This approach models the output weight matrix $\mathbf{W}$ as being drawn from a statistical ensemble and aims to approximate the TCB based on expected values, specifically targeting the root mean square (RMS) norm of the Jacobian. The resulting formula connects $\delta_{\mathrm{TCB}}$ to model parameters like dimensionality and weight variance, as well as higher-order moments of the output probability distribution $o$, offering potentially greater accuracy than simpler scaling laws, particularly when the output distribution is not diffuse (i.e., when $\mathcal{V}_{\mathrm{eff}}$ is small). This statistical perspective contrasts with the exact, non-statistical expression derived in Section G, which applies deterministically to a specific instance of $\mathbf{W}$ and $o$.
The quantity we aim to approximate is:

$$\delta_{\mathrm{TCB}} = \frac{\epsilon}{\|\mathbf{J}_{\mathbf{W}}\|_F},$$

where the key terms involved are:
- $\delta_{\mathrm{TCB}}$: The Token Constraint Bound (Section 2.4), representing a distance in the hidden state space $\mathbf{R}^d$.

- $\epsilon > 0$: A fixed perturbation threshold, representing the target $L_2$ change in the output probability vector $\boldsymbol{o}$. This is typically treated as a dimensionless small value (e.g., 0.01).
- $\mathbf{J_W} \in \mathbb{R}^{\mathcal{V} \times d}$: Jacobian of the softmax output probabilities $\boldsymbol{o}$ with respect to the hidden state $\boldsymbol{h}$. Its Frobenius norm $\|\mathbf{J_W}\|_F$ has units of (probability change) / (hidden state unit).
- $\|\cdot\|_F$: The Frobenius norm of a matrix.
- $\boldsymbol{o} = (o_1, \ldots, o_\mathcal{V})^T \in \mathbf{R}^\mathcal{V}$: The vector of softmax output probabilities, $\sum_{i=1}^{\mathcal{V}} o_i = 1$, $o_i \geq 0$.
- $S_k = \sum_{i=1}^{\mathcal{V}} o_i^k$: The $k$-th moment sum of the output probabilities. Note $S_1 = 1$.
- $\mathcal{V}_{\text{eff}} = 1/S_2$: The effective vocabulary size, a measure of distribution flatness (strictly $\mathcal{V}_{\text{eff}}^{(2)}$).
- $\boldsymbol{h} \in \mathbf{R}^d$: The hidden state vector (embedding dimension $d$).
- $\mathbf{W} \in \mathbf{R}^{\mathcal{V} \times d}$: The output weight matrix (embedding matrix).
- $\sigma^2$: The assumed variance of the elements of $\mathbf{W}$ under the statistical model. This is a parameter of the theoretical model. When applied to a real model, it might represent an empirical estimate of the variance relevant to the specific forward pass or training state.
- $\mathcal{V}$: The size of the vocabulary.

**Step 1: Recap the Jacobian Matrix $\mathbf{J_W}$.** As established in (4), the Jacobian is: Here, the matrix $\mathbf{M} \in \mathbf{R}^{\mathcal{V} \times \mathcal{V}}$ is symmetric and depends only on the output probability vector $\boldsymbol{o}$. It encapsulates how changes in the pre-softmax logits $\boldsymbol{z} = \mathbf{W}\boldsymbol{h}$ translate into changes in the post-softmax probabilities $\boldsymbol{o}$.

**Step 2: Introduce Statistical Assumptions on $\mathbf{W}$.** The core of the statistical approach is to model the output weight matrix $\mathbf{W}$ not as a fixed, trained entity, but as a random matrix drawn from a simple ensemble. We make the following simplifying assumptions about its elements $W_{jk}$:

$$\mathbf{E}[W_{jk}] = 0 \quad \text{and} \quad \mathbf{E}[W_{jk}^2] = \sigma^2 \quad \text{(i.i.d.)}. \tag{12}$$

These imply $\mathbf{E}[W_{jk}W_{lm}] = \sigma^2 \delta_{jl}\delta_{km}$. The zero-mean assumption simplifies calculations. The constant variance $\sigma^2$ captures a typical scale.

**Caveat 1 (Model Simplification):** Realistically trained $\mathbf{W}$ matrices possess significant structure (e.g., semantic clusters, non-zero mean after layer normalization, varying variances per token, correlations between rows $\boldsymbol{w}_i, \boldsymbol{w}_j$). This i.i.d. model ignores such structure. Specifically, it implies $\mathbf{E}[\mathbf{W}\mathbf{W}^\top] = d\sigma^2 \mathbf{I}_\mathcal{V}$, assuming orthogonality between rows on average, which might not hold empirically. For enhanced accuracy, one might incorporate an empirical covariance term $\Sigma_{\text{emp}}$ such that $\mathbf{W}\mathbf{W}^\top \approx d\sigma^2 \mathbf{I} + \Sigma_{\text{emp}}$, though this complicates the derivation.

**Note on $\sigma^2$:** $\sigma^2$ should be interpreted either as a fixed parameter of the theoretical ensemble, or, when connecting to a real model, as an estimate of the empirical variance of weights relevant to the context (e.g., measured during the specific forward pass, possibly after normalization layers).

**Step 3: Approximate the Squared Frobenius Norm via Expectation.** We want to estimate $\|\mathbf{J_W}\|_F^2 = \|\mathbf{M}\mathbf{W}\|_F^2$. Under the statistical model, $\|\mathbf{J_W}\|_F^2$ is a random variable. We approximate it by its expectation $\mathbf{E_W}[\|\mathbf{J_W}\|_F^2]$.

$$\begin{aligned}
\mathbf{E_W}[\|\mathbf{J_W}\|_F^2] &= \mathbf{E_W}\left[\text{Tr}(\mathbf{J_W}\mathbf{J_W}^\top)\right] \\
&= \mathbf{E_W}\left[\text{Tr}(\mathbf{M}\mathbf{W}(\mathbf{M}\mathbf{W})^\top)\right] \\
&= \mathbf{E_W}\left[\text{Tr}(\mathbf{M}\mathbf{W}\mathbf{W}^\top\mathbf{M}^\top)\right] \quad \text{(using } (AB)^\top = B^\top A^\top) \\
&= \mathbf{E_W}\left[\text{Tr}(\mathbf{M}\mathbf{W}\mathbf{W}^\top\mathbf{M})\right] \quad \text{(since } \mathbf{M} \text{ is symmetric)} \tag{13} \\
&= \mathbf{E_W}\left[\sum_{i,j,l=1}^{\mathcal{V}}\sum_{k=1}^{d} M_{ij}W_{jk}W_{lk}M_{li}\right] \quad \text{(Trace expansion)} \tag{14}
\end{aligned}$$

Here we introduce the first major approximation:

**Approximation 1 (Decorrelation):** The matrix elements $M_{ij} = \delta_{ij}o_i - o_i o_j$ depend on $\boldsymbol{o} = \text{softmax}(\mathbf{W}\boldsymbol{h})$, which itself depends on $\mathbf{W}$. Therefore, $\mathbf{M}$ is correlated with $\mathbf{W}$. We make the strong approximation that this correlation can be ignored when computing the expectation involving quadratic terms of $\mathbf{W}$, effectively treating $\mathbf{M}$ as constant with respect to the expectation $\mathbf{E_W}[\cdot]$.

$$\mathbf{E_W}[\|\mathbf{J_W}\|_F^2] \approx \text{Tr}\left(\mathbf{M}\mathbf{E_W}[\mathbf{W}\mathbf{W}^\top]\mathbf{M}\right) \quad \textit{(Approximation 1 Applied)} \tag{15}$$

**Caveat 2 (Decorrelation Risk):** This approximation $\mathbf{E}[MAM] \approx \mathbf{M}\mathbf{E}[A]\mathbf{M}$ can introduce bias. The correlation is likely non-negligible if dimensions $d$ or $\mathcal{V}$ are small, or if the distribution $\boldsymbol{o}$ is highly peaked (low $\mathcal{V}_{\text{eff}}$, where small changes in $\mathbf{W}$ affecting the top logits significantly alter $\mathbf{M}$). The error magnitude requires careful analysis (e.g., via perturbation theory or empirical validation in problematic regimes).

Now, we use the statistical property from (12). The $(j,l)$-th element of $\mathbf{E_W}[\mathbf{WW}^\top]$ is: $(\mathbf{E_W}[\mathbf{WW}^\top])_{jl} = \mathbf{E_W}[(\mathbf{WW}^\top)_{jl}] = \mathbf{E_W}\left[\sum_{k=1}^d W_{jk} W_{lk}\right] = \sum_{k=1}^d \mathbf{E_W}[W_{jk} W_{lk}] = \sum_{k=1}^d \sigma^2 \delta_{jl} = d\sigma^2 \delta_{jl}$. Therefore, $\mathbf{E_W}[\mathbf{WW}^\top] = d\sigma^2 \mathbf{I}_\mathcal{V}$. Substituting this into (15):

$$\begin{aligned}
\mathbf{E_W}[\|\mathbf{J_W}\|_F^2] &\approx \mathrm{Tr}(\mathbf{M}(d\sigma^2 \mathbf{I}_\mathcal{V})\mathbf{M}) \\
&= d\sigma^2 \mathrm{Tr}(\mathbf{M}^2) \quad (\text{Since } \mathbf{M}^T = \mathbf{M}) \\
&= d\sigma^2 \|\mathbf{M}\|_F^2. \quad (\text{Definition of } \|\mathbf{M}\|_F^2 = \mathrm{Tr}(\mathbf{MM}^T))
\end{aligned} \tag{16}$$

This indicates that the expected squared norm of the Jacobian is approximately proportional to the squared norm of the probability-dependent matrix $\mathbf{M}$, scaled by $d\sigma^2$. This connection relies critically on Approximation 1.

**Step 4: Use the Exact Squared Frobenius Norm of M.** The calculation of $\|\mathbf{M}\|_F^2 = \mathrm{Tr}(\mathbf{M}^2)$ is deterministic once $\boldsymbol{o}$ is known. As derived in Section G , the exact value is:

$$\|\mathbf{M}\|_F^2 = \sum_{i=1}^\mathcal{V} o_i^2 (1 - o_i)^2 + \sum_{i \neq j} (o_i o_j)^2 \tag{17}$$

$$= \sum_i (o_i^2 - 2o_i^3 + o_i^4) + \sum_{i \neq j} o_i^2 o_j^2 \tag{18}$$

$$= (S_2 - 2S_3 + S_4) + ((\sum_i o_i^2)(\sum_j o_j^2) - \sum_i (o_i^2)^2) \tag{19}$$

$$= S_2 - 2S_3 + S_4 + S_2^2 - S_4 \tag{20}$$

$$= S_2 - 2S_3 + S_2^2. \tag{21}$$

This expression relates the norm of $\mathbf{M}$ directly to the second ($S_2$) and third ($S_3$) moments of the output probability distribution. Note that this step involves no statistical approximation itself.

**Step 5: Approximate the Jacobian Norm using RMS.** We now introduce the second key approximation:

**Approximation 2 (RMS Substitution):** We approximate the actual Jacobian norm $\|\mathbf{J_W}\|_F$ for a specific $\mathbf{W}$ by its root mean square (RMS) value under the statistical ensemble model:

$$\|\mathbf{J_W}\|_F \approx \sqrt{\mathbf{E_W}[\|\mathbf{J_W}\|_F^2]}. \tag{22}$$

This relies on the assumption that the random variable $\|\mathbf{J_W}\|_F$ concentrates around its RMS value (related to concentration of measure phenomena, plausible for large $d$ or $\mathcal{V}$). However, it remains an approximation. By Jensen's inequality, $\sqrt{\mathbf{E}[X^2]} \geq \mathbf{E}[X]$, so the RMS value typically overestimates the expected norm. The magnitude of this overestimation (and the validity of concentration) depends on the variance of $\|\mathbf{J_W}\|_F^2$, which might be large if the distribution of norms is heavy-tailed (e.g., due to specific weight structures or sparse outputs). More rigorous analysis might require bounding $\mathrm{Var}[\|\mathbf{J_W}\|_F]$ or using concentration inequalities (e.g., Hanson-Wright), which is beyond the scope of this derivation.

Substituting the result for the expected squared norm from (16) and the exact expression for $\|\mathbf{M}\|_F^2$ from (21) into (22):

$$\|\mathbf{J_W}\|_F \approx \sqrt{d\sigma^2 \|\mathbf{M}\|_F^2} = \sqrt{d\sigma^2 (S_2 - 2S_3 + S_2^2)}. \tag{23}$$

This equation provides our refined statistical approximation for the Frobenius norm of the Jacobian.

**Step 6: Refined Approximation for $\delta_{\text{TCB}}$.** Using the definition $\delta_{\text{TCB}} = \epsilon / \|\mathbf{J_W}\|_F$ and substituting the RMS approximation for the norm from (23), we obtain the refined statistical approximation for the Token Constraint Bound:

$$\delta_{\text{TCB}} \approx \frac{\epsilon}{\sqrt{d\sigma^2 (S_2 - 2S_3 + S_2^2)}}. \tag{24}$$

This is the main result of this section. It provides an estimate for $\delta_{\text{TCB}}$ based on the model's embedding dimension ($d$), the assumed variance of its output weights ($\sigma^2$), and the second ($S_2$) and third ($S_3$) moments of its current output probability distribution ($\boldsymbol{o}$).

**Step 7: Connection to the Simpler $\mathcal{V}_{\text{eff}}$-based Approximation.** A commonly cited simpler approximation for $\delta_{\text{TCB}}$ relates it directly to the effective vocabulary size $\mathcal{V}_{\text{eff}} = 1/S_2$, often presented as $\delta_{\text{TCB}} \approx \epsilon\sqrt{\mathcal{V}_{\text{eff}}/(d\sigma^2)}$. We see how this arises from (24) by introducing an *additional* approximation:

**Approximation 3 (Diffuse Distribution):** Assume $\boldsymbol{o}$ is sufficiently diffuse, meaning $p_{\max} = \max_i o_i \ll 1$, corresponding to large effective vocabulary size, $\mathcal{V}_{\text{eff}} \gg 1$. Under this condition, higher moments $S_k$ become small relative to lower moments. As argued in Section J ((51)), if $o_i$ are roughly uniform over $\mathcal{V}_{\text{eff}}$ tokens ($o_i \sim 1/\mathcal{V}_{\text{eff}}$), then $S_3 \approx S_2/\mathcal{V}_{\text{eff}}$ and $S_2^2 \approx S_2/\mathcal{V}_{\text{eff}}$. Thus, for large $\mathcal{V}_{\text{eff}}$, $|-2S_3 + S_2^2| \ll S_2$.

This approximation $S_2 - 2S_3 + S_2^2 \approx S_2$ might fail if the distribution is not sufficiently "uniform-like" even if $\mathcal{V}_{\text{eff}}$ is large (e.g., heavy-tailed distributions where a few moderately high probabilities contribute significantly to $S_3$). Subject to this approximation:

$$S_2 - 2S_3 + S_2^2 \approx S_2 \quad \text{(Approximation 3: Diffuse distribution, } \mathcal{V}_{\text{eff}} \gg 1). \tag{25}$$

Substituting into (24):

$$\delta_{\text{TCB}} \approx \frac{\epsilon}{\sqrt{d\sigma^2 S_2}} = \frac{\epsilon}{\sqrt{d\sigma^2/\mathcal{V}_{\text{eff}}}} = \epsilon\sqrt{\frac{\mathcal{V}_{\text{eff}}}{d\sigma^2}}.$$

This shows the simpler $\mathcal{V}_{\text{eff}}$-based formula is a special case relying on both the statistical model and the diffuse distribution assumption.

## G  EXACT, NON-STATISTICAL EXPRESSION FOR $\|\mathbf{J_W}\|_F$ AND ITS RELATION TO WEIGHTED VARIANCE

This section presents the derivation of the exact mathematical expression for the squared Frobenius norm $\|\mathbf{J_W}\|_F^2$ for a *specific*, given output weight matrix $\mathbf{W} \in \mathbf{R}^{\mathcal{V} \times d}$ and hidden state $\boldsymbol{h} \in \mathbf{R}^d$ (which together determine a specific probability vector $\boldsymbol{o} = \text{softmax}(\mathbf{W}\boldsymbol{h})$). This derivation is purely algebraic and deterministic; it does not rely on any statistical assumptions about $\mathbf{W}$ being drawn from a random ensemble. We establish the correct formula and clarify its relationship to, but distinctness from, the trace of the probability-weighted covariance matrix of the embeddings.
**Notation Recap:**

**Step 1: Squared Frobenius Norm as Sum of Squared Row Norms.** The squared Frobenius norm is the sum of the squared Euclidean norms of its rows. Let $\mathbf{J_W}(i,:) \in \mathbb{R}^{1 \times d}$ denote the $i$-th row of the Jacobian $\mathbf{J_W}$.

$$\|\mathbf{J_W}\|_F^2 = \sum_{i=1}^{\mathcal{V}} \|\mathbf{J_W}(i,:)\|_2^2. \tag{26}$$

**Step 2: Derive the Expression for a Jacobian Row.** We need the components of the $i$-th row, $\mathbf{J_W}(i,:) = [(\mathbf{J_W})_{i1}, \ldots, (\mathbf{J_W})_{id}]$. Recall $\mathbf{J_W} = \boldsymbol{A}(\boldsymbol{o})\mathbf{W}$. The $(i,k)$-th element is:

$$
\begin{aligned}
(\mathbf{J_W})_{ik} &= \sum_{j=1}^{\mathcal{V}} A_{ij} W_{jk} \\
&= \sum_{j=1}^{\mathcal{V}} (\delta_{ij} o_i - o_i o_j) W_{jk} \quad \text{(Definition of } A_{ij}) \\
&= (o_i W_{ik} - o_i^2 W_{ik}) + \sum_{j \neq i} (-o_i o_j) W_{jk} \quad \text{(Splitting sum: } j = i \text{ and } j \neq i) \\
&= o_i W_{ik} - o_i \left( o_i W_{ik} + \sum_{j \neq i} o_j W_{jk} \right) \\
&= o_i W_{ik} - o_i \left( \sum_{j=1}^{\mathcal{V}} o_j W_{jk} \right) \quad \text{(Recombining sum)} \\
&= o_i (W_{ik} - (\sum_{j=1}^{\mathcal{V}} o_j W_{jk})) \\
&= o_i((\boldsymbol{w}_i)_k - (\boldsymbol{\mu_w})_k) = o_i(\boldsymbol{w}_i - \boldsymbol{\mu_w})_k
\end{aligned}
\tag{27}
$$

Here, $W_{jk} = (\boldsymbol{w}_j)_k$ is the $k$-th component of embedding $\boldsymbol{w}_j$, and $(\sum_{j=1}^{\mathcal{V}} o_j W_{jk})$ is the $k$-th component of the mean embedding $\boldsymbol{\mu_w}$. Thus, the entire $i$-th row vector (transposed to match $\boldsymbol{w}_i, \boldsymbol{\mu_w}$ as column vectors) is:

$$
\mathbf{J_W}(i,:)^\top = o_i(\boldsymbol{w}_i - \boldsymbol{\mu_w}).
\tag{28}
$$

This shows that the $i$-th row of the Jacobian represents the deviation of the $i$-th embedding from the mean embedding, scaled by the probability $o_i$.

**Step 3: Substitute Row Norm back into Frobenius Norm Definition.** Using the row expression (28) in the definition (26):

$$
\begin{aligned}
\|\mathbf{J_W}\|_F^2 &= \sum_{i=1}^{\mathcal{V}} \|\mathbf{J_W}(i,:)\|_2^2 \\
&= \sum_{i=1}^{\mathcal{V}} \|o_i(\boldsymbol{w}_i - \boldsymbol{\mu_w})^\top\|_2^2 \\
&= \sum_{i=1}^{\mathcal{V}} o_i^2 \|\boldsymbol{w}_i - \boldsymbol{\mu_w}\|_2^2 \quad \text{(Since } o_i \text{ is a scalar).}
\end{aligned}
\tag{29}
$$

This is the exact, non-statistical expression for the squared Frobenius norm of the Jacobian matrix.

$$
\|\mathbf{J_W}(\boldsymbol{h})\|_F^2 = \sum_{i=1}^{\mathcal{V}} o_i^2 \|\boldsymbol{w}_i - \boldsymbol{\mu_w}\|_2^2
\tag{30}
$$

**Step 4: Introduce the Trace of the Weighted Covariance Matrix.** A related quantity often considered is the trace of the probability-weighted covariance matrix of the embedding vectors: Its trace represents the total variance of the embedding vectors, weighted by the probability distribution

$\boldsymbol{o}$:

$$\mathrm{Tr}[\mathrm{Cov}_{\boldsymbol{o}}\left(\boldsymbol{w}\right)] = \mathrm{Tr}\left[\sum_{i=1}^{\mathcal{V}} o_i(\boldsymbol{w}_i - \boldsymbol{\mu_w})(\boldsymbol{w}_i - \boldsymbol{\mu_w})^\top\right]$$

$$= \sum_{i=1}^{\mathcal{V}} o_i \,\mathrm{Tr}\left[(\boldsymbol{w}_i - \boldsymbol{\mu_w})(\boldsymbol{w}_i - \boldsymbol{\mu_w})^\top\right] \quad \text{(Linearity of trace)}$$

$$= \sum_{i=1}^{\mathcal{V}} o_i(\boldsymbol{w}_i - \boldsymbol{\mu_w})^\top(\boldsymbol{w}_i - \boldsymbol{\mu_w}) \quad \text{(Using } \mathrm{Tr}(\boldsymbol{ab}^\top) = \boldsymbol{b}^\top\boldsymbol{a}\text{)}$$

$$= \sum_{i=1}^{\mathcal{V}} o_i\|\boldsymbol{w}_i - \boldsymbol{\mu_w}\|_2^2. \tag{31}$$

This can also be expressed using the variance identity:

$$\mathrm{Tr}[\mathrm{Cov}_{\boldsymbol{o}}\left(\boldsymbol{w}\right)] = \mathbb{E}_{\boldsymbol{o}}\left[\|\boldsymbol{w}\|_2^2\right] - \|\boldsymbol{\mu_w}\|_2^2 = \left(\sum_{i=1}^{\mathcal{V}} o_i\|\boldsymbol{w}_i\|_2^2\right) - \left\|\sum_{j=1}^{\mathcal{V}} o_j\boldsymbol{w}_j\right\|_2^2. \tag{32}$$

**Step 5: Comparing the Jacobian Norm and the Covariance Trace.** Let us compare the exact squared Jacobian norm (30) with the trace of the covariance matrix (31):

$$\|\mathbf{J_W}\|_F^2 = \sum_{i=1}^{\mathcal{V}} o_i^2\|\boldsymbol{w}_i - \boldsymbol{\mu_w}\|_2^2 \tag{33}$$

$$\mathrm{Tr}[\mathrm{Cov}_{\boldsymbol{o}}\left(\boldsymbol{w}\right)] = \sum_{i=1}^{\mathcal{V}} o_i\|\boldsymbol{w}_i - \boldsymbol{\mu_w}\|_2^2 \tag{34}$$

**Step 6: The Exact Expression for $\delta_{\mathrm{TCB}}$.** Using the correct formula for the squared Jacobian norm (30), the exact Token Constraint Bound is:

$$\delta_{\mathrm{TCB}} = \frac{\epsilon}{\|\mathbf{J_W}\|_F} = \frac{\epsilon}{\sqrt{\sum_{i=1}^{\mathcal{V}} o_i^2\|\boldsymbol{w}_i - \boldsymbol{\mu_w}\|_2^2}}. \tag{35}$$

This is the ground-truth value for $\delta_{\mathrm{TCB}}$ given a specific $\mathbf{W}$ and $\boldsymbol{h}$ (determining $\boldsymbol{o}$).

## H  RELATING THE STATISTICAL APPROXIMATION AND THE EXACT EXPRESSION

This section elucidates the precise relationship between the *refined statistical approximation* for $\|\mathbf{J_W}\|_F$ (derived in Section F ) and the *exact, non-statistical expression* for $\|\mathbf{J_W}\|_F$ (derived in Section G ). Understanding this connection is crucial for interpreting the validity and limitations of the statistical approach and for appreciating why it can serve as a useful, interpretable model despite its simplifying assumptions.

**Recap: The Two Formulas for $\|\mathbf{J_W}\|_F^2$.**
**1. Refined Statistical Approximation ( Section F ):** This approach models $\mathbf{W}$ as a random matrix. The core result approximates the actual squared norm by its expected value under the statistical model:

$$\mathbf{E_W}[\|\mathbf{J_W}\|_F^2] \approx d\sigma^2\|\mathbf{M}\|_F^2 = d\sigma^2(S_2 - 2S_3 + S_2^2), \tag{36}$$

where $\mathbf{M} = \mathrm{diag}(\boldsymbol{o}) - \boldsymbol{oo}^\top$, $\sigma^2$ is the assumed variance of $W_{jk}$, $d$ is the embedding dimension, and $S_k = \sum_i o_i^k$. The final TCB approximation (24) uses the root mean square (RMS) value derived from this expectation: $\|\mathbf{J_W}\|_F \approx \sqrt{\mathbf{E_W}[\|\mathbf{J_W}\|_F^2]}$.
**2. Exact Expression ( Section G ):** For a specific, given $\mathbf{W}$ and $\boldsymbol{o}$, the squared norm is calculated deterministically using the geometry of the embeddings and the probability distribution:

$$\|\mathbf{J_W}\|_F^2 = \mathrm{Tr}[\mathrm{Cov}_{\boldsymbol{o}}\left(\boldsymbol{w}\right)] = \mathbb{E}_{\boldsymbol{o}}\left[\|\boldsymbol{w}\|_2^2\right] - \|\mathbb{E}_{\boldsymbol{o}}\left[\boldsymbol{w}\right]\|_2^2 = \sum_{i=1}^{\mathcal{V}} o_i^2\|\boldsymbol{w}_i - \boldsymbol{\mu_w}\|_2^2, \tag{37}$$

where $\boldsymbol{w}_i$ is the $i$-th embedding vector (row of $\mathbf{W}$), $\boldsymbol{\mu_w} = \mathbb{E}_{\boldsymbol{o}}[\boldsymbol{w}] = \sum_j o_j \boldsymbol{w}_j$, and $\mathrm{Tr}[\mathrm{Cov}_{\boldsymbol{o}}(\boldsymbol{w})]$ is the trace of the probability-weighted covariance matrix of the embeddings. The exact TCB (35) uses the square root of this value.

**The Fundamental Connection: Expectation Bridge.** The theoretical link between these two expressions lies in taking the expectation of the *exact* squared norm (37) over the statistical ensemble assumed for $\mathbf{W}$. When we apply the statistical assumptions ((12)) and the key decorrelation approximation (**Approximation 1** from Section F, see (15)) to the exact formula, we recover the core quantity from the statistical approximation:

$$\mathbf{E_W} \left[ \underbrace{\|\mathbf{J_W}\|_F^2}_{\text{Exact value from (37)}} \right] = \mathbf{E_W} \left[ \mathrm{Tr}(\boldsymbol{A}(\boldsymbol{o}) \mathbf{W} \mathbf{W}^\top \boldsymbol{A}(\boldsymbol{o})) \right] \approx \underbrace{d\sigma^2 \|\mathbf{M}\|_F^2}_{\text{Statistical result (36)}}. \tag{38}$$

Let's briefly trace why this works. The calculation performed in **Step 3** of Section F (starting from $\mathbf{E_W}[\mathrm{Tr}(\mathbf{M} \mathbf{W} \mathbf{W}^\top \mathbf{M})]$) effectively computes the expectation of the exact squared norm, represented in the trace form old. The crucial step is applying **Approximation 1**:

$$\mathbf{E_W}[\mathrm{Tr}(\mathbf{M} \mathbf{W} \mathbf{W}^\top \mathbf{M})] \approx \mathrm{Tr}(\mathbf{M} \mathbf{E_W}[\mathbf{W} \mathbf{W}^\top] \mathbf{M}).$$

This approximation treats $\mathbf{M} = \mathrm{diag}(\boldsymbol{o}) - \boldsymbol{o}\boldsymbol{o}^\top$ as fixed when taking the expectation over $\mathbf{W}$, even though $\boldsymbol{o}$ itself depends on $\mathbf{W}$ via the softmax. Using $\mathbf{E_W}[\mathbf{W}\mathbf{W}^\top]_{ij} = \mathrm{Tr}(\mathbf{E}[\boldsymbol{w}_i \boldsymbol{w}_j^\top]) = \sum_k \mathbf{E}[W_{ik} W_{jk}] = \sum_k \sigma^2 \delta_{ij} \delta_{kk} = d\sigma^2 \delta_{ij}$, we get $\mathbf{E_W}[\mathbf{W}\mathbf{W}^\top] = d\sigma^2 \mathbf{I}_\mathcal{V}$. Substituting this yields:

$$\mathrm{Tr}(\mathbf{M}(d\sigma^2 \mathbf{I}_\mathcal{V})\mathbf{M}) = d\sigma^2 \mathrm{Tr}(\mathbf{M}^2) = d\sigma^2 \|\mathbf{M}\|_F^2.$$

This confirms that $d\sigma^2 \|\mathbf{M}\|_F^2 = d\sigma^2(S_2 - 2S_3 + S_2^2)$ is indeed the result of calculating the expected value of the exact squared norm under the statistical assumptions and the decorrelation approximation.

**Role of Approximations Revisited.** The difference between the final $\delta_{\mathrm{TCB}}$ value predicted by the statistical approximation (24) and the exact value (35) for a specific model arises precisely from the approximations inherent in the statistical derivation:

**Statistical Ensemble Model for W:** The actual trained weight matrix $\mathbf{W}$ is treated as a typical realization from a simple statistical ensemble (i.i.d., zero-mean, variance $\sigma^2$ elements, see (12)). Real trained matrices have complex structure (correlations, non-zero mean, varying variances) not captured by this model. The approximation's accuracy depends on how well the ensemble captures the properties relevant to the norm calculation (specifically, the second moments involved in $\mathbf{W}\mathbf{W}^\top$).

**Decorrelation (M and W):** The calculation in (38) relies on treating the probability-dependent matrix $\mathbf{M}$ as approximately uncorrelated with the quadratic terms in $\mathbf{W}$ (like $\mathbf{W}\mathbf{W}^\top$) when taking the expectation $\mathbf{E_W}$ (**Approximation 1**, (15)). This approximation ignores the feedback loop where $\mathbf{W}$ determines $\boldsymbol{o}$, which in turn determines $\mathbf{M}$. It may be justified by averaging effects in high dimensions $(d, \mathcal{V})$, but it introduces a deviation from the exact expectation.

**RMS Approximation** ($\|\mathbf{J_W}\|_F \approx \sqrt{\mathbf{E}[\|\mathbf{J_W}\|_F^2]}$)**:** The final step in the statistical approximation replaces the actual norm $\|\mathbf{J_W}\|_F$ for the specific $\mathbf{W}$ with the ensemble-averaged RMS value (**Approximation 2**, (22)). This assumes that the norm of the specific matrix is close to the average norm across the ensemble, relying on concentration of measure phenomena. While often plausible for high-dimensional random matrices, the actual norm can deviate from the average.

In summary, the refined statistical TCB approximation (24) estimates the TCB by replacing the exact Jacobian norm $\|\mathbf{J_W}\|_F$ with $\sqrt{d\sigma^2(S_2 - 2S_3 + S_2^2)}$, which represents the approximate RMS norm expected under the simplified statistical model and decorrelation assumption.

## I DERIVATION OF THE EXACT JACOBIAN NORM FORMULA

In this appendix, we provide a detailed step-by-step derivation of the exact mathematical expression for the squared Frobenius norm of the output Jacobian matrix, $\|\mathbf{J_W}(\boldsymbol{h})\|_F^2$. This derivation is performed for a *specific* instance of the output weight matrix $\mathbf{W} \in \mathbb{R}^{\mathcal{V} \times d}$ and the hidden state $\boldsymbol{h} \in \mathbb{R}^d$, which together determine the output probability vector $\boldsymbol{o} = \mathrm{softmax}(\mathbf{W}\boldsymbol{h})$. The derivation relies solely on the definitions of the Jacobian and the Frobenius norm, requiring no statistical assumptions about $\mathbf{W}$.

The final result relates $\|\mathbf{J_W}\|_F^2$ to a weighted sum of squared distances involving the output embedding vectors. We also clarify its relationship to the trace of the probability-weighted covariance matrix of the embeddings, a concept central to related analyses but distinct from the Jacobian norm itself.

**Step 1: Definition of Squared Frobenius Norm via Rows.**  The squared Frobenius norm of any matrix is the sum of the squared Euclidean norms ($L_2$ norms) of its rows. Let $\mathbf{J_W}(i,:)$ denote the $i$-th row vector of the Jacobian matrix $\mathbf{J_W}$.

$$\|\mathbf{J_W}(\boldsymbol{h})\|_F^2 = \sum_{i=1}^{\mathcal{V}} \|\mathbf{J_W}(i,:)\|_2^2. \tag{39}$$

**Step 2: Jacobian Definition and its Relation to Weights.**  Recall the Jacobian matrix is given by $\mathbf{J_W} = \frac{\partial \boldsymbol{o}}{\partial \boldsymbol{h}}$. Using the chain rule, we can write:

$$\mathbf{J_W} = \frac{\partial \boldsymbol{o}}{\partial \boldsymbol{z}} \frac{\partial \boldsymbol{z}}{\partial \boldsymbol{h}} = (\mathrm{diag}(\boldsymbol{o}) - \boldsymbol{o}\boldsymbol{o}^\top)\mathbf{W} = \boldsymbol{A}(\boldsymbol{o})\mathbf{W}.$$

The element $(\mathbf{J_W})_{ik}$ represents $\frac{\partial o_i}{\partial h_k}$.

**Step 3: Explicit Calculation of the $i$-th Row of the Jacobian.**  We aim to find the vector $\mathbf{J_W}(i,:) = [(\mathbf{J_W})_{i1}, (\mathbf{J_W})_{i2}, \ldots, (\mathbf{J_W})_{id}]$. The component $(\mathbf{J_W})_{ik}$ is the $(i,k)$-th element of the matrix product $\boldsymbol{A}(\boldsymbol{o})\mathbf{W}$:

$$
\begin{aligned}
(\mathbf{J_W})_{ik} &= \sum_{j=1}^{\mathcal{V}} A_{ij} W_{jk} \\
&= \sum_{j=1}^{\mathcal{V}} (\delta_{ij} o_i - o_i o_j) W_{jk} \quad \text{(Definition of } A_{ij}) \\
&= (o_i W_{ik}) + \sum_{j \neq i} (-o_i o_j) W_{jk} \\
&= o_i W_{ik} - o_i \sum_{j \neq i} o_j W_{jk} \\
&= o_i W_{ik} - o_i \left( \sum_{j=1}^{\mathcal{V}} o_j W_{jk} - o_i W_{ik} \right) \quad \text{(Completing the sum over } j) \\
&= o_i W_{ik} - o_i \left( \sum_{j=1}^{\mathcal{V}} o_j W_{jk} \right) + o_i^2 W_{ik}
\end{aligned}
$$

Let's restart the derivation for $(\mathbf{J_W})_{ik}$ more directly:

$$
\begin{aligned}
(\mathbf{J_W})_{ik} &= \sum_{j=1}^{\mathcal{V}} A_{ij} W_{jk} \\
&= A_{ii} W_{ik} + \sum_{j \neq i} A_{ij} W_{jk} \quad \text{(Separating diagonal term)} \\
&= o_i(1 - o_i) W_{ik} + \sum_{j \neq i} (-o_i o_j) W_{jk} \quad \text{(Substituting } A_{ij} \text{ values)} \\
&= o_i W_{ik} - o_i^2 W_{ik} - \sum_{j \neq i} o_i o_j W_{jk} \\
&= o_i W_{ik} - o_i \left( o_i W_{ik} + \sum_{j \neq i} o_j W_{jk} \right) \\
&= o_i W_{ik} - o_i \left( \sum_{j=1}^{\mathcal{V}} o_j W_{jk} \right) \quad \text{(Recombining sum)} \tag{40}
\end{aligned}
$$

Now, let's interpret the terms. $W_{jk}$ is the $k$-th component of the $j$-th embedding vector $\boldsymbol{w}_j$. Let $w_{j,k}$ denote this component.

$$\sum_{j=1}^{\mathcal{V}} o_j W_{jk} = \sum_{j=1}^{\mathcal{V}} o_j w_{j,k} = \left(\sum_{j=1}^{\mathcal{V}} o_j \boldsymbol{w}_j\right)_k = (\boldsymbol{\mu_w})_k$$

where $(\boldsymbol{\mu_w})_k$ is the $k$-th component of the mean embedding vector $\boldsymbol{\mu_w}$. Substituting this back into (40):

$$(\mathbf{J_W})_{ik} = o_i W_{ik} - o_i(\boldsymbol{\mu_w})_k = o_i(W_{ik} - (\boldsymbol{\mu_w})_k) = o_i(\boldsymbol{w}_i - \boldsymbol{\mu_w})_k \tag{41}$$

This expression gives the $k$-th component of the $i$-th row of $\mathbf{J_W}$. Therefore, the $i$-th row vector itself is:

$$\mathbf{J_W}(i,:) = o_i(\boldsymbol{w}_i - \boldsymbol{\mu_w})^\top. \tag{42}$$

This shows that each row of the Jacobian is the deviation of the corresponding embedding vector $\boldsymbol{w}_i$ from the mean embedding $\boldsymbol{\mu_w}$, scaled by the probability $o_i$.

**Step 4: Substitute Row Norm back into Frobenius Norm Definition.** Now we substitute the expression for the $i$-th row (42) back into the definition of the squared Frobenius norm (39):

$$
\begin{aligned}
\|\mathbf{J_W}(\boldsymbol{h})\|_F^2 &= \sum_{i=1}^{\mathcal{V}} \|\mathbf{J_W}(i,:)\|_2^2 \\
&= \sum_{i=1}^{\mathcal{V}} \|o_i(\boldsymbol{w}_i - \boldsymbol{\mu_w})^\top\|_2^2 \\
&= \sum_{i=1}^{\mathcal{V}} o_i^2 \|\boldsymbol{w}_i - \boldsymbol{\mu_w}\|_2^2 \quad \text{(Since } o_i \text{ is a scalar).}
\end{aligned} \tag{43}
$$

This equation provides the exact, non-statistical expression for the squared Frobenius norm of the Jacobian. It is determined by the squared distances between each embedding vector $\boldsymbol{w}_i$ and the mean embedding $\boldsymbol{\mu_w}$, weighted by the square of the corresponding probability $o_i^2$.

$$\|\mathbf{J_W}(\boldsymbol{h})\|_F^2 = \sum_{i=1}^{\mathcal{V}} o_i^2 \|\boldsymbol{w}_i - \boldsymbol{\mu_w}\|_2^2 \tag{44}$$

**Step 5: Relationship to the Trace of the Covariance Matrix.** The trace of the probability-weighted covariance matrix of the embedding vectors is a closely related but distinct concept. The covariance matrix is defined as: Its trace is:

$$
\begin{aligned}
\mathrm{Tr}[\mathrm{Cov}_{\boldsymbol{o}}(\boldsymbol{w})] &= \mathrm{Tr}\left[\sum_{i=1}^{\mathcal{V}} o_i(\boldsymbol{w}_i - \boldsymbol{\mu_w})(\boldsymbol{w}_i - \boldsymbol{\mu_w})^\top\right] \\
&= \sum_{i=1}^{\mathcal{V}} o_i \mathrm{Tr}\left[(\boldsymbol{w}_i - \boldsymbol{\mu_w})(\boldsymbol{w}_i - \boldsymbol{\mu_w})^\top\right] \quad \text{(Linearity of trace)} \\
&= \sum_{i=1}^{\mathcal{V}} o_i(\boldsymbol{w}_i - \boldsymbol{\mu_w})^\top(\boldsymbol{w}_i - \boldsymbol{\mu_w}) \quad \text{(Using } \mathrm{Tr}(\boldsymbol{ab}^\top) = \boldsymbol{b}^\top\boldsymbol{a}) \\
&= \sum_{i=1}^{\mathcal{V}} o_i \|\boldsymbol{w}_i - \boldsymbol{\mu_w}\|_2^2.
\end{aligned} \tag{45}
$$

As derived previously, this trace also equals the variance identity:

$$\mathrm{Tr}[\mathrm{Cov}_{\boldsymbol{o}}(\boldsymbol{w})] = \mathbb{E}_{\boldsymbol{o}}\left[\|\boldsymbol{w}\|_2^2\right] - \|\mathbb{E}_{\boldsymbol{o}}[\boldsymbol{w}]\|_2^2 = \left(\sum_{i=1}^{\mathcal{V}} o_i \|\boldsymbol{w}_i\|_2^2\right) - \left\|\sum_{j=1}^{\mathcal{V}} o_j \boldsymbol{w}_j\right\|_2^2. \tag{46}$$

This quantity, $\mathrm{Tr}[\mathrm{Cov}_{\boldsymbol{o}}(\boldsymbol{w})]$, represents the total variance of the embedding vectors, weighted by the probabilities $o_i$.

**Step 6: Comparing Jacobian Norm and Covariance Trace.** Comparing the derived exact squared Jacobian norm (43) with the trace of the covariance matrix (45):

$$\|\mathbf{J_W}\|_F^2 = \sum_{i=1}^{\mathcal{V}} o_i^2 \|\boldsymbol{w}_i - \boldsymbol{\mu_w}\|_2^2 \tag{47}$$

$$\mathrm{Tr}[\mathrm{Cov}_{\boldsymbol{o}}(\boldsymbol{w})] = \sum_{i=1}^{\mathcal{V}} o_i \|\boldsymbol{w}_i - \boldsymbol{\mu_w}\|_2^2 \tag{48}$$

These two quantities are clearly different, differing by a factor of $o_i$ in the weighting term inside the sum. While both measure aspects of the dispersion of the embedding vectors relative to their mean $\boldsymbol{\mu_w}$, they are not mathematically identical. The Jacobian norm gives more weight (via $o_i^2$) to the deviation of high-probability embeddings from the mean.

## J DERIVATION OF REGIME-DEPENDENT TCB BEHAVIOR

In this appendix, we provide detailed mathematical derivations supporting the regime-dependent behavior of the Token Constraint Bound ($\delta_{\mathrm{TCB}}$), as discussed in Section 3 and observed empirically Table 1. We analyze the behavior of $\delta_{\mathrm{TCB}} = \epsilon/\|\mathbf{J_W}(\boldsymbol{h})\|_F$ in two distinct regimes based on the flatness of the output probability distribution $\boldsymbol{o}$: high flatness (large effective vocabulary size, $\mathcal{V}_{\mathrm{eff}} \gg 1$) and low flatness (highly peaked distribution, $\mathcal{V}_{\mathrm{eff}} \approx 1$).

The key is to understand how the squared Frobenius norm of the Jacobian, $\|\mathbf{J_W}(\boldsymbol{h})\|_F^2$, behaves in these limits. We will use different but related expressions for the norm depending on the regime: the statistical approximation for the high-$\mathcal{V}_{\mathrm{eff}}$ regime and the exact formula for the low-$\mathcal{V}_{\mathrm{eff}}$ regime.

**Recap of Key Formulas:**

1. **Exact Squared Norm (from Section I, Eq. (43)):**

$$\|\mathbf{J_W}(\boldsymbol{h})\|_F^2 = \sum_{i=1}^{\mathcal{V}} o_i^2 \|\boldsymbol{w}_i - \boldsymbol{\mu_w}\|_2^2 \tag{49}$$

   where $\boldsymbol{w}_i$ is the $i$-th embedding vector (row of $\mathbf{W}$), $\boldsymbol{o} = (o_1, \ldots, o_\mathcal{V})$ is the probability vector, and $\boldsymbol{\mu_w} = \sum_{j=1}^{\mathcal{V}} o_j \boldsymbol{w}_j$ is the probability-weighted mean embedding.

2. **Refined Statistical Approximation (from Section F, Eq. (23)):**

$$\|\mathbf{J_W}(\boldsymbol{h})\|_F \approx \sqrt{d\sigma^2(S_2 - 2S_3 + S_2^2)} \tag{50}$$

   where $d$ is the hidden dimension, $\sigma^2$ is the assumed variance of weight elements $W_{jk}$, and $S_k = \sum_i o_i^k$ is the $k$-th moment sum ($\mathcal{V}_{\mathrm{eff}} = 1/S_2$). This approximation relies on modeling $\mathbf{W}$ statistically and approximating the norm by its RMS value.

### J.1 REGIME 1: HIGH FLATNESS (LARGE $\mathcal{V}_{\mathrm{eff}} \gg 1$)

**Assumptions.** In this regime, the probability distribution $\boldsymbol{o}$ is spread relatively evenly across many tokens.

- $\mathcal{V}_{\mathrm{eff}} = 1/S_2$ is large.
- Consequently, the maximum probability $p_{\max} = \max_i o_i$ must be small ($p_{\max}\!\ll\!1$). Roughly, if probabilities are spread over $\sim \mathcal{V}_{\mathrm{eff}}$ tokens, then $o_i \sim 1/\mathcal{V}_{\mathrm{eff}}$.
- We use the refined statistical approximation (50), which implicitly assumes the statistical model for $\mathbf{W}$ (i.i.d. elements, zero mean, variance $\sigma^2$) is a reasonable proxy for average behavior.

**Approximating the Jacobian Norm.** We analyze the term $\|\mathbf{M}\|_F^2 = S_2 - 2S_3 + S_2^2$ within the statistical approximation (50). Since $o_i\!\ll\!1$ for all $i$, higher powers of $o_i$ are much smaller. Let's assess the magnitude of the terms relative to $S_2$:

- $S_2 = \sum_i o_i^2 = 1/\mathcal{V}_{\mathrm{eff}}$.
- $S_3 = \sum_i o_i^3 \leq (\max_j o_j) \sum_i o_i^2 = p_{\max} S_2$. If $o_i \sim 1/\mathcal{V}_{\mathrm{eff}}$, then $S_3 \sim \mathcal{V}_{\mathrm{eff}} \times (1/\mathcal{V}_{\mathrm{eff}})^3 = 1/\mathcal{V}_{\mathrm{eff}}^2 = S_2/\mathcal{V}_{\mathrm{eff}}$.
- $S_2^2 = (1/\mathcal{V}_{\mathrm{eff}})^2 = S_2/\mathcal{V}_{\mathrm{eff}}$.

Thus, both $S_3$ and $S_2^2$ are smaller than $S_2$ by a factor of approximately $\mathcal{V}_{\text{eff}}$. Since we assume $\mathcal{V}_{\text{eff}} \gg 1$, the terms $-2S_3$ and $+S_2^2$ become negligible compared to $S_2$:

$$\|\mathbf{M}\|_F^2 = S_2 - 2S_3 + S_2^2 \approx S_2 \quad \text{(for large } \mathcal{V}_{\text{eff}}\text{).} \tag{51}$$

This simplification corresponds to Approximation 3 discussed in Section F . Substituting this back into the statistical norm approximation (50):

$$\|\mathbf{J_W}\|_F \approx \sqrt{d\sigma^2(S_2)} \tag{52}$$

$$= \sqrt{\frac{d\sigma^2}{\mathcal{V}_{\text{eff}}}}. \tag{53}$$

This expression predicts that the Jacobian norm decreases as $\mathcal{V}_{\text{eff}}$ increases.

**Behavior of $\delta_{\text{TCB}}$.** Using the definition $\delta_{\text{TCB}} = \epsilon/\|\mathbf{J_W}\|_F$ and the approximation (53):

$$\delta_{\text{TCB}} \approx \frac{\epsilon}{\sqrt{d\sigma^2/\mathcal{V}_{\text{eff}}}} = \epsilon\sqrt{\frac{\mathcal{V}_{\text{eff}}}{d\sigma^2}}. \tag{54}$$

**Conclusion (High $\mathcal{V}_{\text{eff}}$):** In the high-flatness regime, $\delta_{\text{TCB}}$ is predicted to be approximately proportional to the square root of the effective vocabulary size:

$$\delta_{\text{TCB}} \propto \sqrt{\mathcal{V}_{\text{eff}}} \quad \text{(for } \mathcal{V}_{\text{eff}} \gg 1\text{)}$$

This provides a clear mathematical basis for the strong positive correlation $r_{\delta,\mathcal{V}_{\text{eff}}} \approx 0.95$ observed empirically ( Table 1 ) in diverse datasets where distributions are often flat.

**Negligible Influence of Margin $z_k - z_{j^*}$.** The margin $z_k - z_{j^*} = z_k - z_{j^*}$ between the logits of two specific tokens $k$ (usually the top prediction) and $j^*$ (usually the top competitor) directly affects $o_k$ and $o_{j^*}$. In the high-$\mathcal{V}_{\text{eff}}$ regime, however, both $o_k$ and $o_{j^*}$ are typically small (e.g., $\sim 1/\mathcal{V}_{\text{eff}}$). Changes in $z_k - z_{j^*}$ primarily redistribute a small amount of probability mass between these two (and possibly nearby) tokens. The impact on the overall sum $S_2 = \sum o_i^2$, which aggregates contributions from many small probabilities, is minimal. Consequently, changes in $z_k - z_{j^*}$ have a very weak effect on $\|\mathbf{J_W}\|_F$ via (53), and therefore also on $\delta_{\text{TCB}}$. This explains the near-zero correlation $r_{\delta,z_k-z_{j^*}}$ observed in diverse datasets ( Table 1 ).

## J.2 REGIME 2: LOW FLATNESS (SMALL $\mathcal{V}_{\text{eff}} \approx 1$)

**Assumptions.** In this regime, the probability distribution $\boldsymbol{o}$ is sharply peaked on a single token.
- $\mathcal{V}_{\text{eff}} \approx 1$. This occurs when one probability, say $o_k$, is close to 1.
- Let $o_k = 1 - \epsilon_s$, where $\epsilon_s = \sum_{j \neq k} o_j$ is a small positive quantity ($\epsilon_s \ll 1$).
- All other probabilities $o_j$ ($j \neq k$) are very small, typically $o_j \sim O(\epsilon_s)$ or smaller.
- We use the exact expression for the norm (49) as it directly captures the influence of the specific dominant embedding $\boldsymbol{w}_k$ and its relation to competitors. Statistical averaging inherent in (50) is less appropriate here.

**Approximating the Jacobian Norm.** We analyze the exact sum $\|\mathbf{J_W}\|_F^2 = \sum_{i=1}^{\mathcal{V}} o_i^2\|\boldsymbol{w}_i - \boldsymbol{\mu_w}\|_2^2$. First, approximate the mean embedding $\boldsymbol{\mu_w}$:

$$\boldsymbol{\mu_w} = \sum_{j=1}^{\mathcal{V}} o_j\boldsymbol{w}_j = o_k\boldsymbol{w}_k + \sum_{j \neq k} o_j\boldsymbol{w}_j \tag{55}$$

$$= (1 - \epsilon_s)\boldsymbol{w}_k + \sum_{j \neq k} o_j\boldsymbol{w}_j \tag{56}$$

$$= \boldsymbol{w}_k - \epsilon_s\boldsymbol{w}_k + \sum_{j \neq k} o_j\boldsymbol{w}_j \tag{57}$$

$$= \boldsymbol{w}_k + \sum_{j \neq k} o_j(\boldsymbol{w}_j - \boldsymbol{w}_k) \quad \text{(using } \epsilon_s = \sum_{j \neq k} o_j\text{)} \tag{58}$$

This shows $\boldsymbol{\mu_w}$ is close to $\boldsymbol{w}_k$, differing by terms of order $O(\epsilon_s)$.
Now, consider the terms in the sum for $\|\mathbf{J_W}\|_F^2$:

- **Term for $i = k$:** We need $\|\boldsymbol{w}_k - \boldsymbol{\mu_w}\|_2^2$. From (58):

$$\boldsymbol{w}_k - \boldsymbol{\mu_w} = -\sum_{j \neq k} o_j(\boldsymbol{w}_j - \boldsymbol{w}_k)$$

The squared norm is $\|\boldsymbol{w}_k - \boldsymbol{\mu_w}\|_2^2 = \left\|\sum_{j \neq k} o_j(\boldsymbol{w}_j - \boldsymbol{w}_k)\right\|_2^2$. Since each $o_j = O(\epsilon_s)$ for $j \neq k$, this squared norm is $O(\epsilon_s^2)$. The contribution to the total sum is $o_k^2\|\boldsymbol{w}_k - \boldsymbol{\mu_w}\|_2^2 \approx (1-\epsilon_s)^2 O(\epsilon_s^2) \approx O(\epsilon_s^2)$. This term is therefore negligible to the leading order.

- **Terms for $i \neq k$:** We need $\|\boldsymbol{w}_i - \boldsymbol{\mu_w}\|_2^2$. Using (58):

$$\boldsymbol{w}_i - \boldsymbol{\mu_w} = \boldsymbol{w}_i - \left(\boldsymbol{w}_k + \sum_{j \neq k} o_j(\boldsymbol{w}_j - \boldsymbol{w}_k)\right)$$

$$= (\boldsymbol{w}_i - \boldsymbol{w}_k) - \sum_{j \neq k} o_j(\boldsymbol{w}_j - \boldsymbol{w}_k)$$

The squared norm is:

$$\|\boldsymbol{w}_i - \boldsymbol{\mu_w}\|_2^2 = \left\|(\boldsymbol{w}_i - \boldsymbol{w}_k) - \sum_{j \neq k} o_j(\boldsymbol{w}_j - \boldsymbol{w}_k)\right\|_2^2$$

Expanding the square:

$$\|\boldsymbol{w}_i - \boldsymbol{\mu_w}\|_2^2 = \|\boldsymbol{w}_i - \boldsymbol{w}_k\|_2^2 - 2(\boldsymbol{w}_i - \boldsymbol{w}_k)^\top \left(\sum_{j \neq k} o_j(\boldsymbol{w}_j - \boldsymbol{w}_k)\right) + \left\|\sum_{j \neq k} o_j(\boldsymbol{w}_j - \boldsymbol{w}_k)\right\|_2^2$$

The middle term is $O(\epsilon_s)$, and the last term is $O(\epsilon_s^2)$. Thus, to leading order:

$$\|\boldsymbol{w}_i - \boldsymbol{\mu_w}\|_2^2 \approx \|\boldsymbol{w}_i - \boldsymbol{w}_k\|_2^2 + O(\epsilon_s) \quad \text{(for } i \neq k\text{)}$$

The contribution of the $i$-th term ($i \neq k$) to the total sum $\|\mathbf{J_W}\|_F^2$ is $o_i^2\|\boldsymbol{w}_i - \boldsymbol{\mu_w}\|_2^2$. Since $o_i^2 = O(\epsilon_s^2)$, the contribution is:

$$o_i^2\|\boldsymbol{w}_i - \boldsymbol{\mu_w}\|_2^2 \approx o_i^2(\|\boldsymbol{w}_i - \boldsymbol{w}_k\|_2^2 + O(\epsilon_s)) = o_i^2\|\boldsymbol{w}_i - \boldsymbol{w}_k\|_2^2 + O(\epsilon_s^3)$$

Summing the contributions for all $i$:

$$\|\mathbf{J_W}\|_F^2 = \underbrace{o_k^2\|\boldsymbol{w}_k - \boldsymbol{\mu_w}\|_2^2}_{\approx O(\epsilon_s^2)} + \sum_{j \neq k} \underbrace{o_j^2\|\boldsymbol{w}_j - \boldsymbol{\mu_w}\|_2^2}_{\approx o_j^2\|\boldsymbol{w}_j - \boldsymbol{w}_k\|_2^2 + O(\epsilon_s^3)}$$

$$\approx \sum_{j \neq k} o_j^2\|\boldsymbol{w}_j - \boldsymbol{w}_k\|_2^2 \quad \text{(keeping leading order terms, } O(\epsilon_s^2)\text{)} \tag{59}$$

This approximation reveals that for highly peaked distributions, the squared Jacobian norm is dominated by the sum of squared distances between the dominant embedding $\boldsymbol{w}_k$ and competitor embeddings $\boldsymbol{w}_j$, weighted by the *square* of the competitors' small probabilities $o_j^2$.

**Connecting to Margin $z_k - z_{j^*}$.** The logit margin between the winning token $k$ and any other token $j$ is $z_k - z_{j^*} = z_k - z_j$. The probability $o_j$ for $j \neq k$ can be approximated using the softmax definition when $z_k$ is large compared to others:

$$o_j = \frac{e^{z_j}}{\sum_{l=1}^{\mathcal{V}} e^{z_l}} = \frac{e^{z_j}}{e^{z_k} + \sum_{l \neq k} e^{z_l}} \approx \frac{e^{z_j}}{e^{z_k}(1 + \sum_{l \neq k} e^{z_l - z_k})} \approx \frac{e^{z_j}}{e^{z_k}} = e^{-(z_k - z_j)} = e^{-z_k - z_{j^*}}$$

This approximation holds because $\sum_{l \neq k} e^{z_l - z_k} = \sum_{l \neq k} o_l/o_k \approx \epsilon_s/(1 - \epsilon_s) \approx \epsilon_s \ll 1$. Substituting this into the norm approximation (59):

$$\|\mathbf{J_W}\|_F^2 \approx \sum_{j \neq k} (e^{-z_k - z_{j^*}})^2 \|\boldsymbol{w}_j - \boldsymbol{w}_k\|_2^2 = \sum_{j \neq k} e^{-2z_k - z_{j^*}} \|\boldsymbol{w}_j - \boldsymbol{w}_k\|_2^2 \tag{60}$$

The specific margin defined as $z_k - z_{j^*} = z_k - z_{j^*}$, where $j^*$ is the top competitor (highest logit $z_j$ among $j \neq k$), corresponds to the term with the largest $e^{-2z_k - z_{j^*}}$ (smallest $z_k - z_{j^*}$) in the sum, which often dominates the sum's value.

**Behavior of $\delta_{\text{TCB}}$.** As the margin $z_k - z_{j^*}$ increases, the corresponding $z_k - z_{j^*}$ for the closest competitors also increases. This leads to an exponential decrease in the terms $e^{-2z_k - z_{j^*}}$ in (60). Consequently, $\|\mathbf{J_W}\|_F^2$ decreases strongly (exponentially) as $z_k - z_{j^*}$ increases. Since $\delta_{\text{TCB}} = \epsilon/\|\mathbf{J_W}\|_F$, an increase in $z_k - z_{j^*}$ causes a decrease in the denominator, leading to an *increase* in $\delta_{\text{TCB}}$. **Conclusion (Low $\mathcal{V}_{\text{eff}}$):** In the low-flatness regime, TCB increases rapidly as the logit margin $z_k - z_{j^*}$ increases:

$$\delta_{\text{TCB}} \propto \frac{1}{\sqrt{\sum_{j \neq k} e^{-2(z_k - z_j)} \|\boldsymbol{w}_j - \boldsymbol{w}_k\|_2^2}} \approx \text{Increases strongly with } z_k - z_{j^*} \quad (\text{for } \mathcal{V}_{\text{eff}} \approx 1)$$

This derivation provides the theoretical underpinning for the strong positive correlation $r_{\delta, z_k - z_{j^*}} \approx 0.62$ observed empirically in the low-$\mathcal{V}_{\text{eff}}$ data subset ( Table 1 ).

**Negligible Influence of Residual $\mathcal{V}_{\text{eff}}$ Variation.** In this regime, $\mathcal{V}_{\text{eff}}$ is already close to 1. Small changes in the probability distribution (e.g., caused by changing $z_k - z_{j^*}$) lead to minuscule changes in $\mathcal{V}_{\text{eff}}$. Specifically, $\mathcal{V}_{\text{eff}} = 1/S_2 = 1/(o_k^2 + \sum_{j \neq k} o_j^2)$. As $o_k \approx 1$ and $o_j = O(\epsilon_s)$, we have $\mathcal{V}_{\text{eff}} \approx 1/(1 - 2\epsilon_s + O(\epsilon_s^2))$. While changes in $z_k - z_{j^*}$ affect $\epsilon_s$ and thus cause small fluctuations in $\mathcal{V}_{\text{eff}}$, these variations are vastly outweighed by the direct exponential impact of $z_k - z_{j^*}$ on $\|\mathbf{J_W}\|_F^2$ via (60). This explains why the correlation $r_{\delta, \mathcal{V}_{\text{eff}}}$ drops to nearly zero ($\approx 0.08$) in the low-$\mathcal{V}_{\text{eff}}$ regime.

