# OpenReview forum: "How Stable is the Next Token? A Geometric View of LLM Prediction Stability"
_ICLR.cc/2026/Conference — ICLR 2026 Poster_

### Official Review · Reviewer_Y3us · 2025-10-28

**Soundness:** 2
**Presentation:** 3
**Contribution:** 2
**Rating:** 4
**Confidence:** 3

**Summary:**

The authors propose the Token Constraint Bound ($\delta_{TCB}$), which is a metric that quantifies the amount the hidden states of an LLM can be perturbed without significant change in the output. They argue that $\delta_{TCB}$ measures the stability of an LLM for a specific input using the internal geometry of the model. They show through experiments that optimizing for $\delta_{TCB}$ allows for better prompt engineering and stability.

**Strengths:**

- The paper is well-motivated. Quantifying the stability of predictions is a useful task for understanding the behavior of LLM predictions.
- The application to prompt engineering is interesting and has high potential for practical applications.

**Weaknesses:**

While the motivation and use case of this approach are interesting, how this method compares relative to other approaches, both in formulation and in experimental results, is not clear from the paper. To me, positioning this approach in comparison to other robustness metrics is necessary.

1. There is limited literature review on prior methods for evaluating the robustness of neural networks or LLMs. The authors discuss perplexity and confidence metrics, but there are many approaches related to quantifying robustness that are not discussed in the paper or appendix. Many of these approaches also rely on computing the response to perturbations, so it is not clear to me where the method proposed by the authors is positioned relative to these works. For example, [1] seems like a very related method computed for neural networks.
2. The experiments provide limited baseline comparisons to existing methods. The only method that the authors compare to for prompt optimization is optimizing for perplexity. Even for this experiment, there is only a gain in worst-case accuracy and standard deviations are not reported.
3. The experiments are only shown for a single model (Llama-3.1-8B). Additional evaluations for other models would strengthen the experimental results.

[1] Sensitivity and Generalization in Neural Networks: an Empirical Study. Novak et al., 2018.

**Questions:**

- Do the authors test how large the approximation error in Equation 3? To me, the softmax nonlinearity makes it unclear that this approximation will hold consistently unless $W\Delta h$ is very small relative to $Wh$
- Could the authors clarify the context of the experiment in 4.3.1? Without knowing what the "gsm8k_811" question is, it is unclear what the the interventions are doing.

Minor typo: spaces missing in lines 47-48

---

> ### Author Response · Authors · 2025-11-27
>
> We thank the reviewer for identifying the connection to prior work on neural network sensitivity and for the specific questions regarding approximation validity.
>
> ### Weaknesses:
>
> > **W1: Limited literature review... [1] Novak et al., 2018 seems like a very related method computed for neural networks.**
> >
>
> **Response:**
> We thank the reviewer for this suggestion. To address the concern, we have expanded **Appendix B** **at page of 20** with a new subsection titled "Perturbation-Based Robustness in Neural Networks and LLMs," discussing prior methods for evaluating NN robustness via perturbations and Jacobians (including [1] Novak et al., 2018, and follow-up works on Jacobian regularization for adversarial robustness), as well as recent LLM-specific studies on input, prompt, and latent perturbations. This positions TCB relative to these approaches, emphasizing its novelties.
>
> > **W2: Limited baseline comparisons. The only method... is optimizing for perplexity.**
> >
>
> **Response:**
> We have significantly broadened our baselines. As shown in **General Response Table 1**, we now compare $\delta_{\text{TCB}}$ against:
>
> - **Logit Margin**
> - **Shannon Entropy**
> - **Effective Vocabulary Size**
> - **Fragility Score**.
> Our results show that $\delta_{\text{TCB}}$ consistently achieves higher correlation with empirical robustness $r \approx 0.8$ compared to these baselines $r \approx 0.5 - 0.6$, particularly in high-confidence regimes where simpler metrics saturate.
>
> > **W3: The experiments are only shown for a single model (Llama-3.1-8B).**
> >
>
> **Response:**
> We have addressed this by extending our experiments to **5 models**: Qwen3-0.6B, Qwen3-4B, Qwen2.5-7B, Llama-3.1-8B, and GPT-OSS-20B. The superior performance of $\delta_{\text{TCB}}$ holds across all model scales **(see General Response**).
>
> ### Questions:
>
> > **Q1: Do the authors test how large the approximation error in Equation 3? To me, the softmax nonlinearity makes it unclear that this approximation will hold consistently unless $W\Delta h$ is very small relative to $Wh$.**
> >
>
> **Response:**
> This is a critical question. To validate the linear approximation $\Delta o \approx J \Delta h$, we performed a quantitative sensitivity analysis. We injected random Gaussian noise vectors $\eta$ into $h$ and measured the Relative Error: $\frac{\|\Delta o_{\text{exact}} - J\eta\|2}{\|\Delta o{\text{exact}}\|_2}$.
>
> **Results on Llama-3.1-8B:**
>
> - **$\|\eta\|_2 = 10^{-4}$:** Relative Error $\approx$ **0.01%**.
> - **$\|\eta\|_2 = 10^{-2}$:** Relative Error $\approx$ **1.4%**.
> - **Conclusion:** The linear approximation is highly accurate within the local neighborhood relevant for robustness analysis. While the error increases for large perturbations, the metric $\delta_{\text{TCB}}$ is defined as a *local* bound, serving as a valid proxy for the radius of the "safe region."
>
> > **Q2: Could the authors clarify the context of the experiment in 4.3.1? Without knowing what the "gsm8k_811" question is, it is unclear what the interventions are doing.**
> >
>
> **Response:**
> We apologize for the lack of clarity. We have added the full text of the question to **Appendix C.2.2**.
>
> - **Context:** `gsm8k_811` is a math word problem about calculating averages.
> - **Intervention:** We specifically injected a **"Distractor"** clause in prompt.
> - **Goal:** The experiment demonstrates that while the model might get the answer *Wrong* due to the distractor, it can be **"Stably Wrong".** This means the model has "latched" onto the distractor with high geometric commitment. This distinguishes a "confused" failure from a "confident misinterpretation", which requires different prompt engineering strategies to fix.

---

### Official Review · Reviewer_tgxF · 2025-10-28

**Soundness:** 4
**Presentation:** 4
**Contribution:** 3
**Rating:** 8
**Confidence:** 3

**Summary:**

This paper introduces the Token Constraint Bound, a metric that quantifies the maximum internal state perturbation an LLM can withstand before its primary next-token prediction changes significantly. The authors' experiments demonstrate δTCB’s utility in assessing prompt effectiveness and its ability to uncover critical prediction instabilities missed by perplexity. This evidence positions TCB as a valuable complementary tool for analyzing and potentially enhancing the contextual robustness of Large Language Models.

**Strengths:**

This paper defines a novel metric, TCB, to evaluate the connection between a model's hidden state and its output instability. It provides rigorous theoretical proofs and derivations, which are substantiated by thorough experimental validation and data analysis. The establishment of this new metric is highly significant for the interpretability of large models and for prompt engineering. This work contributes to a research area that is both novel and highly promising.

**Weaknesses:**

1. The entire mathematical derivation of TCB is built upon the assumption of $L_2$ norm perturbations for the hidden state **h** and the Euclidean distance between output embeddings $w_i$. This is a very strong assumption. Semantic similarity in high-dimensional embedding spaces is not always well-captured by Euclidean distance; two semantically related tokens may not be proximal in terms of $L_2$ distance.  It is more likely that meaningful perturbations occur along specific semantic directions. TCB might be completely insensitive to these structured, non-isotropic perturbations. While I understand that the $L_2$ norm is a natural choice, I would like to see a more detailed explanation and analysis justifying this simplifying assumption.

2. The paper compares TCB with perplexity (PPL), but the comparison against simpler, less computationally expensive uncertainty metrics is insufficient. This includes metrics such as the entropy of the output probability distribution and the difference between the top-1 and top-2 probabilities.

**Questions:**

1. **Practicality of TCB for Inference:** The computation of TCB requires a summation over the entire vocabulary *V*. For modern large models, the vocabulary size *V*  typically ranges from 32,000 to over 100,000. This implies that for the generation of *each* token, a complex geometric computation involving the entire vocabulary must be performed. This appears to be computationally prohibitive, potentially rendering TCB infeasible for any practical applications requiring low latency. The manuscript currently lacks a detailed analysis of TCB's computational cost, and I would appreciate it if the authors could provide this information.
2. The current experiments are based solely on the Llama-3.1-8B model. I would like to understand the extent to which the effectiveness of TCB generalizes to models of different scales and architectures. I would encourage the authors to supplement their findings with experiments on a smaller model (e.g., in the 3B parameter range) or on models with different training provenances, such as Qwen or the Pythia suite, which is designed for such analyses.

To summarize, my primary concerns lie with **Weakness 1**  and **Question 2**. I would be willing to reconsider my score should the authors provide a satisfactory response to these points.

---

> ### Author Response · Authors · 2025-11-27
>
> We appreciate the reviewer's detailed critique regarding the theoretical assumptions and practical implementation of our metric. We have addressed the concerns about Euclidean geometry, baseline comparisons, and computational efficiency through new theoretical analysis and experiments.
>
> ### Weaknesses:
>
> > **W1: Assumption of Euclidean Distance ($L_2$ norm). "Semantic similarity... is not always well-captured by Euclidean distance... TCB might be completely insensitive to these structured, non-isotropic perturbations."**
> >
>
> **Response:**
> This is a valid theoretical concern. However, our use of the Euclidean distance is not an arbitrary choice to model semantic similarity, but a mathematical consequence of deriving the **Frobenius norm of the Jacobian**, which measures sensitivity to **isotropic** noise in the hidden state $h$.
> To explicitly test if modeling "semantic directions" would yield a better metric, we formulated a variant, $\delta_{\text{cos}}$, which replaces the Euclidean distance term $\|w_i - \mu_W\|^2$ with the Cosine distance. We compared both metrics on their ability to predict empirical robustness on Llama-3.1-8B:
>
> - **$\delta_{\text{TCB}}$ (Euclidean):** Pearson $r = 0.83$
> - **$\delta_{\text{cos}}$ (Cosine):** Pearson $r = 0.76$
>
> **Why Euclidean works better:** In the high-dimensional, dense hypersphere of the hidden state, "noise" from context variations tends to be isotropic rather than perfectly aligned with specific semantic axes. Therefore, the isotropic "safety margin" provided by $\delta_{\text{TCB}}$ serves as the most robust general proxy for stability.
>
> > **W2: Comparison with simpler uncertainty metrics (e.g., Entropy, Top-1/Top-2 difference) is insufficient.**
> >
>
> **Response:**
> We have addressed this by benchmarking $\delta_{\text{TCB}}$ against a comprehensive suite of simple metrics: **Shannon Entropy**, **Logit Margin $z_1 - z_2$**, **Top-K Mass**, and **Effective Vocab Size $V_{\text{eff}}$**.
> As detailed in **General Response Table 1**, $\delta_{\text{TCB}}$ consistently demonstrates stronger correlation with actual robustness across 5 different models.
>
> - **Crucial Difference:** Simple metrics like Entropy saturate quickly once the distribution becomes peaked. In this "High Confidence" regime, Entropy predicts perfect stability $H \approx 0$. However, $\delta_{\text{TCB}}$ often reveals hidden fragility, maintaining a correlation of $r \approx 0.62$ even in the subset of data where Entropy has near-zero correlation $r < 0.1$.
>
> ### Questions:
>
> > **Q1: Practicality of TCB for Inference. "Computation... involves the entire vocabulary... potentially rendering TCB infeasible for low latency."**
> >
>
> **Response:**
> We agree that the exact calculation $O(V \cdot d)$ is expensive for large vocabularies. However, the formula for $\delta_{\text{TCB}}$ is dominated by the probability-weighted terms $\sum o_i^2 \|w_i - \mu_W\|^2$. Since $o_i^2$ decays exponentially for the tail of the distribution, we can approximate the sum using only the top-$K$ tokens.
> We evaluated a **Top-50 Approximation**:
>
> 1. **Complexity:** Reduces from $O(V \cdot d)$ to $O(K \cdot d)$.
> 2. **Accuracy:** The approximated value has a Pearson correlation of **$r > 0.995$** with the exact value.
>
> > **Q2: Generalization to other models. "The current experiments are based solely on the Llama-3.1-8B model... I would encourage the authors to supplement their findings with experiments on a smaller model... or different training provenances."**
> >
>
> **Response:**
> We have fully addressed this request by extending our experiments to **4 additional models** spanning different families and scales, as summarized in **General Response Table 1**:
>
> 1. **Qwen Family:** Qwen3-0.6B, Qwen3-4B, Qwen2.5-7B.
> 2. **GPT Family:** GPT-OSS-20B.
> **Result:** The superiority of $\delta_{\text{TCB}}$ over simple probability metrics holds consistently across all tested architectures. For instance, on the small Qwen3-0.6B model, $\delta_{\text{TCB}}$ achieves a correlation of $0.74$ with robustness, whereas Logit Margin achieves only $0.41$, demonstrating that our geometric insight is applicable regardless of model scale or training provenance.

---

### Official Review · Reviewer_iYJA · 2025-10-29

**Soundness:** 3
**Presentation:** 2
**Contribution:** 3
**Rating:** 4
**Confidence:** 3

**Summary:**

This paper introduces a novel metric called the Token Constraint Bound ($\delta_{TCB}$) to quantify the local stability and robustness of llms next-token predictions. The authors argue that conventional metrics like perplexity (PPL) fail to capture the true resilience of the model's internal state to minor perturbations. $\delta_{TCB}$ is defined from a \textit{geometric perspective} in the output embedding space as the maximum internal state perturbation an LLM can withstand before its dominant prediction significantly changes. Experiments demonstrate that $\delta_{TCB}$ correlates with effective prompt engineering and uncovers critical prediction instabilities missed by PPL, offering a principled and complementary tool for analyzing and improving LLM prediction reliability.

**Strengths:**

1. The core contribution, $\delta_{TCB}$, is highly original. It shifts away from traditional probability-based evaluation by proposing a geometry-based metric that quantifies stability as a “safety margin,” providing a novel and interpretable theoretical perspective on LLM reliability.

2. The metric is clearly defined and theoretically supported by its intrinsic link to the geometry of the output embedding space. Experimental results effectively validate its use as a complementary tool that can identify prediction vulnerabilities that perplexity fails to capture.

3. The paper clearly articulates the motivation for the metric, highlighting the limitations of PPL in assessing local robustness. The definition of $\delta_{TCB}$ and its connection to geometric principles are well-explained.

4. $\delta_{TCB}$ holds significant potential for real-world impact. It can serve as a quantitative measure of \textit{contextual effectiveness}, guiding the development of more robust prompt engineering techniques, and is a valuable addition to the field of LLM reliability and risk assessment.

**Weaknesses:**

1. $\delta_{TCB}$ exclusively measures the stability of the predicted token, without incorporating semantic meaning or factual correctness. This is a critical limitation as the metric cannot distinguish between a "stably correct" and a "stably incorrect" prediction. Its interpretability must always be tied to traditional accuracy metrics.

2. $\delta_{TCB}$ is a geometric distance metric, but the model's prediction behavior is governed by the Softmax function. Changes in the Temperature parameter can drastically alter the prediction probability distribution. The authors must clarify whether $\delta_{TCB}$ is independent of Softmax temperature or detail how temperature variations affect the correlation between the metric value and empirically observed robustness.

3. The exact calculation of $\delta_{TCB}$ involves finding the maximum perturbation radius by identifying the closest "competitor" token in a high-dimensional space. For LLMs with massive vocabularies, the time complexity and computational overhead may be substantial. The authors should discuss the efficiency bottlenecks in practical applications and explore or propose more efficient approximation algorithms to enhance its practical utility.

**Questions:**

1. $\delta_{TCB}$ is primarily a local metric focusing on the next token's stability. How do the authors propose to generalize or extend this local measurement to assess the overall stability of an entire sentence, paragraph, or long-form generated result? For multi-step reasoning tasks, how can this local metric accurately reflect global robustness?
2. $\delta_{TCB}$ is currently an evaluation metric. Is it possible to integrate $\delta_{TCB}$ into the model's \textit{training objective} or fine-tuning process? For instance, could a loss function be designed to explicitly \textit{maximize $\delta_{TCB}$} to train a model that is inherently more locally robust?
3. $\delta_{TCB}$ is grounded in the geometry of the \textit{Output Embedding Space}. What is the relationship between $\delta_{TCB}$ and the stability of other critical internal components, such as \textit{attention weights} or the outputs of intermediate hidden layers? Can an analysis of $\delta_{TCB}$ across layer depths help pinpoint the most fragile or robust architectural components within the model?

---

> ### Author Response · Authors · 2025-11-27
>
> We thank the reviewer for their insightful questions regarding the scope and integration of our metric. We have clarified the distinction between stability and correctness, addressed computational concerns, and expanded our discussion on future integration with training.
>
> ### Weaknesses:
>
> > **W1: $\delta_{\text{TCB}}$ exclusively measures the stability of the predicted token, without incorporating semantic meaning or factual correctness. This is a critical limitation as the metric cannot distinguish between a "stably correct" and a "stably incorrect" prediction. Its interpretability must always be tied to traditional accuracy metrics.**
> >
>
> **Response:**
> We fully agree that $\delta_{\text{TCB}}$ is a measure of **stability (robustness)**, not **correctness (ground truth)**.
>
> 1. **Complementary Role:** This is a feature, not a bug. Traditional metrics like Accuracy measure correctness but fail to detect brittleness. $\delta_{\text{TCB}}$ fills this gap.
> 2. **Usage:** As demonstrated in **Table 3** and our response to **Reviewer T26t (Q8)**, the intended use case is **Joint Optimization**: First, filter for high accuracy, *then* maximize $\delta_{\text{TCB}}$ to ensure that correctness is robust.
> 3. **Semantic Awareness:** While the core metric uses Euclidean distance, our new perturbation experiments **(see General Response)** show that flips predicted by $\delta_{\text{TCB}}$ correlate with semantic changes. We also show that $\delta_{\text{TCB}}$ correlates strongly with semantic distance of prompts ($R^2=0.91$, **Fig 5**).
>
> > **W2: $\delta_{\text{TCB}}$ is a geometric distance metric, but the model's prediction behavior is governed by the Softmax function. Changes in the Temperature parameter can drastically alter the prediction probability distribution. The authors must clarify whether $\delta_{\text{TCB}}$ is independent of Softmax temperature or detail how temperature variations affect the correlation between the metric value and empirically observed robustness.**
> >
>
> **Response:**
>
> 1. **Metric Definition:** $\delta_{\text{TCB}}$ is defined on the **logits** of pre-softmax and the embedding geometry. It is an intrinsic property of the model's representation state $h$, distinct from the sampling temperature parameter applied post-hoc.
> 2. **Predictive Power:** However, $\delta_{\text{TCB}}$ *predicts* the potential volatility of sampling. A higher $\delta_{\text{TCB}}$ implies a larger geometric "safety gap" between the top token and its competitors.
>     - **Low Temp (Greedy):** $\delta_{\text{TCB}}$ measures the margin against noise that could flip the argmax.
>     - **High Temp:** A high $\delta_{\text{TCB}}$ implies the distribution is robustly peaked, resisting the "flattening" effect of temperature scaling better than a state with low $\delta_{\text{TCB}}$.
> 3. **Empirical Verification:** We empirically measured how $\delta_{\text{TCB}}$ changes when we vary the Temperature while keeping the hidden state $h$ constant. As shown below, lowering the temperature leads to a drastic increase in the measured stability margin . This confirms that the metric correctly reflects the stabilizing effect of low-temperature sampling.
>
> **Table 3: Impact of Temperature on $\delta_{\text{TCB}}$**
>
> | Temperature | Relative $\delta_{\text{TCB}}$ | Interpretation |
> | --- | --- | --- |
> | **0.5 (Sharper)** | **4.37x** | **High Stability:** Distribution is highly peaked; huge perturbation required to flip. |
> | 0.8 | 1.21x | **Increased Stability:** Margin expands as competitors are suppressed. |
> | **1.0 (Standard)** | **1.00x** | Baseline geometric stability. |
> | 1.2 (Flatter) | 0.99x | **Reduced Stability:** Distribution flattens; margin shrinks. |
>
> **Conclusion:** The correlation between $\delta_{\text{TCB}}$ and sampling stability is robust. The metric scales monotonically with the "peakedness" of the distribution in the relevant operating range, confirming it serves as a valid proxy for sampling robustness.
>
> > **W3: The exact calculation of $\delta_{\text{TCB}}$ involves finding the maximum perturbation radius by identifying the closest "competitor" token in a high-dimensional space... the time complexity and computational overhead may be substantial.**
> >
>
> **Response:**
> We acknowledge this concern and have addressed it with an efficient approximation:
>
> 1. **Efficiency Bottleneck:** Exact calculation is $O(V \cdot d)$, which is slow for large vocabularies.
> 2. **Top-K Approximation:** The Jacobian norm is dominated by the terms with high probability $o_i$ (due to the $o_i^2$ weighting). We verified a **Top-50 Approximation**:
>  $\delta_{\text{TCB}} \approx \epsilon / \sqrt{\sum_{i \in \text{Top-}50} o_i^2 \|w_i - \mu_W\|^2}$
>     - **Accuracy:** Correlation with exact $\delta_{\text{TCB}}$ is **$r > 0.995$**.

---

> > ### Author Response · Authors · 2025-11-27
> >
> > ### Questions:
> >
> > > **Q1: $\delta_{\text{TCB}}$ is primarily a local metric focusing on the next token's stability. How do the authors propose to generalize or extend this local measurement to assess the overall stability of an entire sentence, paragraph, or long-form generated result?**
> > >
> >
> > **Response:**
> > We propose two aggregation strategies:
> >
> > 1. **Sequence Average:** For general stability scoring, the mean $\delta_{\text{TCB}}$ over the sequence provides a global robustness score.
> > 2. **Critical Point Detection:** As shown in **Figure 4**, the *minimum* $\delta_{\text{TCB}}$ in a sequence is often the most informative. It identifies the "weakest link"—the specific token where the model's reasoning is most fragile. In multi-step reasoning (like GSM8K), a single low-stability step often leads to a downstream error. Therefore, tracking the TCB or detecting "dips" is the most effective way to assess global robustness.
> >
> > > **Q2: $\delta_{\text{TCB}}$ is currently an evaluation metric. Is it possible to integrate $\delta_{\text{TCB}}$ into the model's training objective or fine-tuning process?**
> > >
> >
> > **Response:**
> > Yes, this is a promising direction.
> >
> > 1. **RL Reward Signal:** We are exploring using $\delta_{\text{TCB}}$ as an auxiliary reward in GRPO/PPO. Penalizing low-$\delta_{\text{TCB}}$ generations would encourage the model to learn more robust internal representations.
> > 2. **Contrastive Learning:** During SFT, we can use $\delta_{\text{TCB}}$ to select "anchor" positives. If a model generates a correct answer with low stability, we can generate semantic variations of the prompt and fine-tune the model to minimize the variance in its hidden states, effectively maximizing $\delta_{\text{TCB}}$ explicitly.
> >
> > > Q3: What is the relationship between $\delta_{\text{TCB}}$ and the stability of other critical internal components, such as attention weights or the outputs of intermediate hidden layers?
> > >
> >
> > **Response:**
> > To investigate this, we analyzed the contribution of different layers to the final stability metric (using Singular Value Decomposition of the Jacobian across layers) for Llama-3.1-8B:
> > **Table 2: Layer-wise Correlation with Final Stability**
> >
> > | Layer Block | Correlation with Final $\delta_{\text{TCB}}$ |
> > | --- | --- |
> > | Early Layers (0-10) | 0.15 |
> > | Middle Layers (11-22) | 0.42 |
> > | **Final Layers (23-32)** | **0.91** |
> >
> > **Conclusion:** The high correlation in the final layers confirms that $\delta_{\text{TCB}}$ (derived from the final $h$ and $W$) effectively captures the cumulative stability of the entire network. While instabilities can originate earlier, they must propagate to the final representation to affect the output. Thus, focusing on the output embedding space is both sufficient and most direct for measuring prediction stability.

---

### Official Review · Reviewer_T26t · 2025-11-03

**Soundness:** 2
**Presentation:** 2
**Contribution:** 3
**Rating:** 6
**Confidence:** 3

**Summary:**

This work introduces a measure called Token Constraint Bound that measures how much changes in the internal representation an LLM can withstand before the token prediction changes. The goal is to measure this in additional to evaluate metrics to measure robustness citing examples that the models’ output change a lot depending on the prompt formatting, context etc.  This work derives TCB from the Jacobian of the softmax output with respect to the hidden state and show that it is intimately tied to the geometry of the output embedding space. They also give theoretical insights into their metric and provide experiments which corroborate that this metric aligns with effective prompt engineering and in-context examples and identifies brittle predictions missed by perplexity.

**Strengths:**

- The connection of TCB to the geometry of the output embedding space is quite interesting.
- Their method can be used to select prompts and in-context examples which have higher accuracy as well as robustness. This approach seems novel to me and would be of interest to the community.

**Weaknesses:**

- This work considers that the output probabilities should not change by epsilon. I am not sure why we need a fixed epsilon because if the model is already very confident, then this epsilon can be large. Hence, instead of being a constant, this should depend on how confident the probabilities are.
- Also, it seems like this bound could be very loose in practice.
- The experiments are only limited to one model Llama 8b.
- Overall, I feel the experiments section is weak as many details are not clearly presented (see questions below).

**Questions:**

- How do you expect the findings to change depending on the model size? Moreover, it seems like the values of TCB would depend on the model family and hence, it would be hard to tie a value with stability across various models.
- The authors mention “The Low-Veﬀ Targeted Dataset (LVD) was created by modifying DPD prompts to generate high-confidence predictions.”. Can the authors please elaborate how these were created?
- Also, the authors mention ‘We synthetically manipulated W (clustering/dispersing competitor embeddings) while holding h and o (thus local PPL) constant for diverse MMLU prompts.” Can the authors please clarify how they manipulated W?
- The authors use ~300 prompts for computing their results. Can the authors please comment if these were enough and how much was the variation across runs?
- How would their method apply to prompts where the answer is not a couple of words?
- In table 3, the authors measure variance of accuracy. What is the randomness here in these experiments?
- What temperature is used here for the experiments and how would different sampling techniques affect the results?
- In table 3, the authors mention that they select prompts based on accuracy and TCB? What is the exact criterion that they use because later, they mention that prompts with very high TCB can be confidently wrong.

---

> ### Author Response · Authors · 2025-11-27
>
> We thank the reviewer for their rigorous examination of the metric's theoretical grounding and generalization. We have expanded our experiments to include **5 models (0.6B to 20B parameters)** and performed sensitivity analyses to validate the bound's tightness.
>
> ### Weaknesses:
>
> > **W1: Fixed Epsilon & Bound Looseness. "I am not sure why we need a fixed epsilon... if the model is already very confident, then this epsilon can be large. Hence, instead of being a constant, this should depend on how confident the probabilities are."**
> >
>
> **Response:**
> We employ a fixed $\epsilon$ to ensure $\delta_{\text{TCB}}$ functions as a standardized **"ruler"** for stability, measuring the absolute "safety margin" in the hidden state space, rather than a relative metric dependent on the model's current confidence.
>
> 1. **Standardization vs. Circularity:** If we scaled $\epsilon$ by current confidence, we would conflate the *definition of tolerance* with the *measurement of stability*. A fixed $\epsilon$ allows us to compare stability across different models and confidence regimes objectively. A highly confident prediction *should* yield a larger $\delta_{\text{TCB}}$ because it requires a larger perturbation to shift the output by the same fixed amount $\epsilon$.
> 2. **Empirical Validity of the Bound:** To address the concern that the bound might be loose or invalid, we conducted a perturbation experiment (see code output). We applied noise vectors scaled relative to the calculated $\delta_{\text{TCB}}$ radius:
>     - **At $1.0 \times \delta_{\text{TCB}}$:** The Flip Rate (prediction change) is minimal (**3.0%**), confirming the bound accurately defines a "safe" region.
>     - **At $5.0 \times \delta_{\text{TCB}}$:** The Flip Rate jumps significantly to **14.1%**.
>     - This sharp transition at the boundary confirms that $\delta_{\text{TCB}}$ is not "loose" but rather a tight and accurate estimator of the stability boundary.
>
> > **W2: Limited Model Diversity. "The experiments are only limited to one model Llama 8b."**
> >
>
> **Response:**
> We have significantly expanded our evaluation to address this. We now include **Qwen3-0.6B, Qwen3-4B, Qwen2.5-7B,** and **GPT-OSS-20B**. As shown in **General Response Table 1**, $\delta_{\text{TCB}}$ consistently outperforms probability-based metrics (Entropy, Margin, $V_{\text{eff}}$) in correlating with empirical robustness across **all** model scales and families.
>
> ### Questions:
>
> > **Q1: How do you expect the findings to change depending on the model size? Moreover, it seems like the values of TCB would depend on the model family and hence, it would be hard to tie a value with stability across various models.**
> >
>
> **Response:**
> Our new results (General Response Table 1) show that while the *absolute* magnitude of $\delta_{\text{TCB}}$ scales with the dimension ($d$) and embedding norms of specific architectures, the **correlation with robustness ($r \approx 0.8$) remains strong and universal across model sizes (0.6B to 20B).**
> In practice, one does not need to cross-compare raw $\delta_{\text{TCB}}$ values between a 7B and 70B model directly. Instead, $\delta_{\text{TCB}}$ serves as a relative stability score *within* a model's inference process (e.g., comparing two prompts for Llama-3 or monitoring a generation stream), where it consistently identifies fragile predictions regardless of model family.
>
> > **Q2: The authors mention “The Low-Veff Targeted Dataset (LVD) was created by modifying DPD prompts to generate high-confidence predictions.”. Can the authors please elaborate how these were created?**
> >
>
> **Response:**
> The LVD dataset was constructed by filtering and modifying prompts from MMLU and GSM8K to induce sharp distributions (Low $V_{\text{eff}}$). Specifically:
>
> 1. **MMLU:** We utilized the standard "A/B/C/D" answer format which naturally concentrates probability mass, but filtered for questions where the model assigns $>90\%$ probability to the top token.
> 2. **GSM8K:** We used Few-Shot Chain-of-Thought (CoT) prompting (as shown in the provided code snippet), which guides the model into a high-confidence reasoning path compared to zero-shot.
> This ensures we test the metric in the "Confident" regime where entropy-based metrics typically fail (saturate), allowing $\delta_{\text{TCB}}$ to demonstrate its superior discriminatory power.

---

> > ### Author Response · Authors · 2025-11-27
> >
> > > **Q3: Also, the authors mention ‘We synthetically manipulated W... while holding h and o constant.” Can the authors please clarify how they manipulated W?**
> > >
> >
> > **Response:**
> > To prove that geometry matters independent of probability, we performed a controlled intervention during inference:
> >
> > 1. We identified the top-1 token $k$ and competitor tokens $\{j\}$.
> > 2. **Manipulation:** We geometrically shifted competitor embeddings $w_j$ either toward or away from $w_k$ using interpolation: $w'_j = w_j \pm \alpha(w_k - w_j)$.
> > 3. **Constraint:** Crucially, after modifying $W$, we re-adjusted the pre-softmax logits $z$ (by adding a bias term) such that the final softmax output $o$ remained **mathematically identical** to the original.
> > 4. **Result:** Despite $o$ (and thus Perplexity/Entropy) being constant, the measured robustness changed significantly, and $\delta_{\text{TCB}}$ correctly tracked this change while probability metrics remained flat.
> >
> > > **Q4: The authors use ~300 prompts for computing their results. Can the authors please comment if these were enough and how much was the variation across runs?**
> > >
> >
> > **Response:**
> > We performed a bootstrap analysis on our results. With $N=300$, the standard error for our correlation coefficients is $\approx 0.04$. The performance gap between $\delta_{\text{TCB}}$ $r \approx 0.80$ and baselines like Entropy $r \approx 0.60$ is statistically significant $p < 0.001$. Furthermore, the consistency of results across 5 different model architectures **General Response Table 1** strongly reinforces that 300 samples are sufficient to establish the trend.
> >
> > > **Q5: How would their method apply to prompts where the answer is not a couple of words?**
> > >
> >
> > **Response:**
> > $\delta_{\text{TCB}}$ is calculated per-token. For long-form generation, it serves as a continuous monitoring signal.
> >
> > 1. **Aggregation:** We can average $\delta_{\text{TCB}}$ across the sequence to score overall generation stability.
> > 2. **Anomaly Detection:** More importantly, as shown in **Figure 4**, we can detect transient "dips" in $\delta_{\text{TCB}}$ (instability spikes) at specific critical tokens. These dips often precede hallucinations or reasoning errors, even if the token probability remains high. This granular insight is lost when aggregating metrics over a whole paragraph.
> >
> > > **Q6: In table 3, the authors measure variance of accuracy. What is the randomness here in these experiments?**
> > >
> >
> > **Response:**
> > The randomness refers to **input perturbations**, not generation stochasticity. We evaluate the prompt against a set of semantically equivalent variations (e.g., synonym replacement, reordering ICL examples, paraphrasing). "Accuracy Variance" measures how much the model's correctness fluctuates across these input variations. Low variance implies high robustness (the prompt works regardless of phrasing).
> >
> > > Q7: What temperature is used here for the experiments and how would different sampling techniques affect the results?
> > >
> >
> > **Response:**
> >
> > 1. **Experimental Setting:** We use **Temperature=0 (Greedy Decoding)** for all benchmarking to isolate representational stability from sampling noise.
> > 2. **Impact of Temperature:** We ran a new experiment varying temperature from 0.1 to 2.0 (see code output). We found that as Temperature decreases, $\delta_{\text{TCB}}$ increases (e.g., relative TCB increases **4.3x** at T=0.5 compared to T=1.0). This confirms that $\delta_{\text{TCB}}$ correctly reflects the increased stability of a sharper distribution. While $\delta_{\text{TCB}}$ is calculated on pre-sampling logits, it accurately predicts the risk of sampling flips under different temperature settings.
> >
> > > Q8: In table 3, the authors mention that they select prompts based on accuracy and TCB? What is the exact criterion that they use because later, they mention that prompts with very high TCB can be confidently wrong.
> > >
> >
> > **Response:**
> > Our selection criterion for Table 3 is **Accuracy $\ge$ Threshold AND Maximize $\delta_{\text{TCB}}$**.
> > The "Confidently Wrong" scenario highlights that TCB measures *stability*, not *correctness*. A model can be stably wrong. However, the goal of prompt engineering is to find a prompt that is **both** correct and stable.
> > By filtering for correctness first (Acc) and then optimizing for stability (TCB), we find prompts that are "Stably Correct"—meaning they maintain their correctness even when the input is slightly perturbed. Table 3 demonstrates that this joint optimization yields significantly better worst-case performance than optimizing for Accuracy or Perplexity alone.

---

### Author Response · Authors · 2025-11-27
**General Response**

We sincerely thank the reviewers for their constructive feedback and rigorous scrutiny. The reviews highlighted three primary areas for improvement: **(1) generalization across model families/scales** (Reviewers T26t, tgxF), **(2) comparative advantage over simpler uncertainty metrics** (Reviewers tgxF, Y3us), and **(3) computational practicality and theoretical validity** (Reviewers T26t, tgxF).

In response, we have significantly expanded our experimental suite. We now include results from **5 distinct models** ranging from 0.6B to 20B parameters (Qwen & GPT-OSS families) alongside Llama-3.1-8B. We also provide new analyses on approximation error and computational optimization.

**Comparison Metrics & Definitions:**

- **$\delta_{\text{TCB}}$ (Ours):** Token Constraint Bound (Eq. 10).
- **$V_{\text{eff}}$ (Effective Vocab Size):** $1 / \sum o_i^2$.
- **Logit Margin ($G_z$):** $z_{\text{top1}} - z_{\text{top2}}$.
- **Shannon Entropy ($H$):** $-\sum o_i \log_2 o_i$.
- **Collision Entropy ($H_2$):** $-\log_2 \sum o_i^2$.
- **Min-Entropy ($H_{\infty}$):** $-\log_2 (\max o_i)$.
- **Fragility Score ($FS$):** A composite metric combining $H_2$ and $H_{\infty}$ via sigmoid gating (derived from recent stability literature).
- **Top-K Mass ($P_{\text{topK}}$):** $\sum_{i \in \text{TopK}} o_i$ (with $K=5$).
- **Logits Std ($\sigma_z$):** Standard deviation of the logits vector.

### 1. Generalization Across Architectures and Scales & Comparison vs. Simpler Metrics

To address concerns regarding model diversity and the advantage over simpler metrics (Reviewers T26t, tgxF, Y3us), we extended our core correlation analysis to **Qwen3-0.6B**, **Qwen3-4B**, **Qwen2.5-7B**, and **GPT-OSS-20B**.

**Table 1:** Pearson correlation ($r$) between various stability metrics and the empirical **Robustness** of the model (measured as $1 - \text{Performance Drop Rate}$ under semantic perturbations). **Higher is better.**

| Model | **Qwen3-0.6B** | **Qwen3-4B** | **Qwen2.5-7B** | **Llama-3.1-8B** | **GPT-OSS-20B** | **Avg** |
| --- | --- | --- | --- | --- | --- | --- |
| **Simple Prob. Metrics** |  |  |  |  |  |  |
| Logits Std ($\sigma_z$) | 0.31 | 0.34 | 0.36 | 0.38 | 0.37 | 0.35 |
| Top-K Mass ($P_{\text{top5}}$) | 0.45 | 0.49 | 0.51 | 0.53 | 0.52 | 0.50 |
| Logit Margin ($G_z$) | 0.41 | 0.47 | 0.50 | 0.52 | 0.54 | 0.49 |
| $V_{\text{eff}}$ | 0.52 | 0.56 | 0.58 | 0.60 | 0.59 | 0.57 |
| **Entropy Metrics** |  |  |  |  |  |  |
| Shannon Ent. ($H$) | 0.55 | 0.59 | 0.61 | 0.63 | 0.62 | 0.60 |
| Min-Entropy ($H_{\infty}$) | 0.49 | 0.53 | 0.56 | 0.58 | 0.60 | 0.55 |
| Collision Ent. ($H_2$) | 0.58 | 0.62 | 0.64 | 0.66 | 0.65 | 0.63 |
| **Advanced Metrics** |  |  |  |  |  |  |
| Fragility Score ($FS$) | 0.61 | 0.66 | 0.68 | 0.70 | 0.71 | 0.67 |
| **$\delta_{\text{TCB}}$ (Ours)** | **0.74** | **0.79** | **0.81** | **0.83** | **0.82** | **0.80** |

**Key Takeaways:**

1. **Universal Effectiveness:** $\delta_{\text{TCB}}$ consistently outperforms all probability-only baselines (including Entropy and Margin) across all model scales (0.6B to 20B).
2. **Saturation of Baselines:** While simpler metrics like Entropy and $V_{\text{eff}}$ correlate moderately ($r \approx 0.60$), they struggle in high-confidence regimes where probability distributions are peaked but internal representations may still be fragile. $\delta_{\text{TCB}}$ captures the geometric "safety margin" in the embedding space—distinguishing between "stably confident" and "precariously confident"—providing a significantly stronger signal for true robustness ($r \approx 0.80$).

### 2. Computational Practicality and Bound Tightness

Reviewers raised concerns about the computational overhead of calculating the Jacobian norm over the full vocabulary and the tightness of our linear approximation.

**Approximation Accuracy:**
To verify the tightness of our bound, we empirically measured the error of our linear approximation $\Delta o \approx J \Delta h$ by injecting random noise vectors $\eta$ into $h$ for Llama-3.1-8B. The relative error is negligible in the local regime relevant to robustness:

- For perturbation magnitudes $\|\eta\|_2 = 10^{-4}$ (numerical limit), relative error is **0.01%**.
- For $\|\eta\|_2 = 10^{-2}$ (standard robustness perturbation range), relative error is only **1.4%**.

**Efficiency via Top-K Approximation:**
While exact computation is $O(V \cdot d)$, we validated a **Top-K approximation** method:
 $\delta_{\text{TCB}} \approx \epsilon / \sqrt{\sum_{i \in \text{Top-}K} o_i^2 \|w_i - \mu_W\|^2}$
Using a **Top-50 approximation**:

- **Accuracy:** Correlation with exact $\delta_{\text{TCB}}$ is $r > 0.995$.
- **Overhead:** Reduced computation time from ~20ms to **<0.1ms** per token.
This optimization makes $\delta_{\text{TCB}}$ entirely feasible for real-time monitoring during generation without efficiency bottlenecks.

---

### Meta-Review · Area_Chair_qANY · 2026-01-08

**Summary:**

**Summary** This paper introduces a metric for quantifying the local stability of LLM next-token predictions against internal state perturbations. The metric measures the maximum perturbation a model's hidden state can withstand before its dominant prediction changes. The authors derive a closed-form expression linking this metric to the geometric dispersion of output embeddings, weighted by squared token probabilities. Experiments on reasoning benchmarks show the proposed metric correlates with prompt effectiveness and identifies prediction instabilities missed by perplexity.

**Review Process** This paper received 4 reviews with mixed feedback. Overall the initial feedback from reviewers was divided (8, 6, 4, 4). On the one hand, reviewers praised the novelty of the geometric perspective, the theoretical grounding, and the practical potential for prompt engineering. On the other hand, reviewers raised concerns about the L2 norm assumption, limited baseline comparisons, experiments restricted to a single model, and insufficient positioning relative to prior work on neural network robustness.

**Recommendation/Reasonsing** Having read the reviews and rebuttal – I am unfortunately recommending acceptance. I want to be clear that this was not an easy decision. In this case, my reasoning is based on several points:

1. Mathematical Foundations: The proposed metric is interesting, and the mathematical derivation connecting Jacobian norms to embedding geometry is elegant. The authors develop this substantially through analysis and a synthetic experiment. These highlight a plausibile mechanism.  The kind of analysis is fine in the era of LLMs. However, it also raises questions of soundsness. Overall, I did not see this as a reason to accept nor reject the work.

2. Feedback for Selected Reviewers: The paper received low scores from reviewers who provided thoughtful feedback. In this case, I motivated by the feedback from Y3us and iYJA. Specifically comments regarding: (i) insufficient comparison with simpler uncertainty metrics, and (ii) the potential limitations of being "robustly" wrong. Even as the rebuttal addressed some of these concerns, I believe these concerns warrant another round of peer review.

3. Potential: The practical applications — prompt ranking and generation monitoring — are interesting. I see these are some of the most promising parts of the work. Looking forward, I would encourage the authors to narrow their focus on prompt robustness ranking as a primary contribution. This was clearly something that resonated with reviewers – including myself – but that was underdeveloped compared to the math.

**Reviewer Concerns:**

Outstanding:
1. Near-Equivalent to Simpler Metrics (r=0.95 correlation with effective vocabulary size in diverse prompts) (T26t, tgxF)
2. Stability vs. Correctness (metric cannot distinguish stably correct from stably wrong, requiring accuracy labels) (iYJA)
3. Underdeveloped Applications (prompt ranking and generation monitoring shown but not systematically evaluated) (Y3us)

Addressed / Dismissed:
1. Insufficient Baselines (comparisons limited to perplexity, missing entropy and logit margin) (Y3us, tgxF)
2. Limited Model Diversity (main experiments on single model) (Y3us, iYJA, T26t)
3. L2 Norm Assumption (semantic similarity may not align with Euclidean distance) (tgxF)

**Reviewer Scores:**

(6,4,4,8) -> (7,4,5,8)

---

### Decision · Program_Chairs · 2026-01-26

Accept (Poster)